# Damping analysis of Floating Offshore Wind Turbine (FOWT): a new control strategy reducing the platform vibrations

Matteo Capaldo[1] and Paul Mella[2]

[1]TotalEnergies OneTech, Palaiseau, France
[1,2]Ecole Polytechnique, Institut Polytechnique de Paris, Palaiseau, France

**Correspondence:** Matteo Capaldo (matteo.capaldo@totalenergies.com)

**Abstract.** In this paper, the coupled dynamics of the floating platform and the wind turbine rotor are analysed. In particular, the damping is explicitly derived from the coupled equations of rotor and floating platform. The analysis of the damping leads to the study of the instability phenomena obtaining the explicit conditions that lead to the Non Minimum Phase Zero (NMPZ). Two NMPZs are analysed, one related to the rotor dynamics and the other one to the platform pitch dynamics. The latter introduces a novelty and an explicit condition is provided in this work for its verification. In the second part of the paper, from the analysis of the damping of the floating platform, a new strategy for the control of Floating Offshore Wind Turbines (FOWTs) is proposed. This strategy allows one to impose to the controller an explicit level of damping in the platform pitch motion that adapts with wind speed and operating conditions without changing the period of platform pitching. Finally the new strategy is compared to one without compensation and one with a non-adapting compensation by performing aero-hydro-servo-elastic numerical simulations of a reference FOWT. Generated power, motions, blade pitch and tower base fatigue are compared showing that the new control strategy can reduce fatigue in the structure without affecting the power production.

## 1 Introduction

Wind energy is an important source of renewable energy and it has a very high potential both onshore and offshore. In terms of installed capacity, onshore wind is still the largest contribution. However, in the next years, the new annual offshore installed capacity is estimated to exceed 30 GW by 2030, in order to stay on-track for a netzero/$1.5°\,C$ pathway (Lee et al., 2022). In offshore, there is a growing interest in floating foundations. In fact, FOWTs would allow access to good wind resource locations that are not suitable for fixed-bottom foundations.

In that context, the levelized cost of energy (LCOE) of offshore wind farms needs to be decreased to be competitive with respect to onshore wind. This is especially true for the FOWTs. One effective way to achieve this objective is to investigate different strategies for the control of the FOWTs. As explained in (Bianchi et al., 2007), the minimisation of the LCOE involves a series of partial objectives, energy capture, mechanical loads and power quality. These objectives are actually closely related and sometimes conflicting and they should not be pursued separately. Hence, the question is to find a well balanced compromise among them. Considering FOWTs, this optimization problem increases in complexity since the motions of the floating platform

interact with the feedback control loop. Moreover, the coupling between the platform motions, the rotor dynamics and the blade

pitch control can lead to oscillating ( not damped) steady-state or even to unstable conditions (Larsen et al., 2007).

Those phenomena can be classified in two families: one is related to undesired motions of the platform, even if the system is still stable. Those are the Non Minimum Phase Zeros (NMPZs). They are associated to the zeros of the transfer function describing the system. The other family is associated to the damping of the system which is related to the poles of the transfer function and can affect the system stability.

The nature and the set of control parameters leading to those phenomena can vary from one platform design to another one, e.g., barge, spar, semi-sub or tension-leg platform. However, for each of the platforms, there exist sets of design and control parameters leading to undesired behaviours (Fleming et al., 2014).

Bottom-fixed control strategies normally consider a squared law for the electrical torque control (below-rated wind speeds) and a set of integral and proportional coefficients (the pitch scheduling) to control the rotational speed by the blade pitch and

35 operate the wind turbine at the desired steady-state conditions (above-rated wind speeds) (Lopez-Queija et al., 2022). This control strategy allows to operate the wind turbine in steady-state conditions for a large set of wind speeds, typically from 3 to $25\ ms^{-1}$.

To adapt this control strategy to FOWTs, a first compensation is considered in this work. It aims at solving the NMPZ effects caused by the blade pitch on the rotor rotational dynamics. This solution is already introduced in (Fischer, 2013) (Stockhouse

et al., 2021) and it is, in this work, analytically developed. The study of the NMPZs brings to a new NMPZ phenomenon, described for the first time in this work. This is the NMPZ caused by the blade pitch on the platform dynamics. This new phenomenon is analytically developed leading to the explicit condition to verify it. However, the compensations proposed in literature and adopted in this work can't correct it.

A second compensation considered in this work aims at solving the issue of the coupling between the platform motions

and the rotor dynamics leading to non-damped oscillations of the systems. This phenomenon can be even amplified when the bottom-fixed pitch control is considered for a floating wind turbine. The issue comes from the fact that in above-rated wind speed, the blade pitch regulates the speed by increasing the angle of attack to feather. For a FOWT, when the platform has a forward motion, the rotor experiences an increasing wind speed. It means an increasing aerodynamic torque which tends to accelerate the rotor. Consequently, the blade pitch control increases the angle of attack to feather and, hence, reduces the

aerodynamic torque and regulates the rotor speed. However, it also reduces the rotor thrust that induces a further forward motion. So the blade pitch control amplifies the original forward motion of the platform because the floating platform surge and pitch natural frequencies are in the bandwidth of the blade pitch controller.

Solutions exist to avoid this phenomenon. The first one and the most common in literature is to reduce the blade pitch control proportional and integral gains in order to reduce its bandwidth and exclude the platform pitch and surge natural frequencies

(Jonkman et al., 2008; Larsen et al., 2007). However, this solution does not completely solve the problem and, moreover, the price to pay is to have a less reactive blade pitch control that allows important over-speeds of the rotor. Alternative methods use additional sensing, such as nacelle fore-aft acceleration measurements or platform gyroscopes, to improve the performance of the pitch controller. In (Abbas et al., 2022), authors introduce a correction of the blade pitch control proportional to the platform

pitch velocity in order to decouple the rotor dynamics from the platform pitch motions. An explicit form for the compensating parameter is proposed to obtain this decoupling, considering the first order linear expression of the rotor dynamics variation with respect to the platform pitch. Here, we also propose a correction of the blade pitch control proportional to the platform pitch velocity. However, differently from (Abbas et al., 2022), we propose to take advantage of the coupling between platform dynamics and rotor dynamics in order to define an explicit value of the platform pitch damping, obtained by compensating the blade pitch. The two strategies arrive to different expressions of the proportional coefficients. This difference leads to coefficients with opposite signs.

The control strategy proposed in this work shares some similarities to the ones introduced in (Lackner et al., 2009) and (Lenfest et al., 2020). In (Lenfest et al., 2020) the platform pitch damping and the compensation parameter are investigated with a purely numerical approach. Here, we propose a mathematical frame and an explicit formulation for the compensation parameter related to this damping which depends on the system properties. As introduced in (Lackner et al., 2009), we also use the platform pitch velocity to adjust the rated speed set-point in order to reduce platform motions. The rated generator speed is no longer a constant value but a function of the platform pitch velocity and the blade pitch is used to damp the floating platform pitch. However, differently from (Lackner et al., 2009), the ratio between proportional and integral gains of the correction can be considered different for the platform pitch motion and the rotor speed.

Higher order controllers, such as a linear quadratic regulator (LQR) are applied and evaluated in (Ma et al., 2018). A disturbance accommodating controller (DAC) is evaluated in (Menezes et al., 2018) and it is coupled with individual pitch control (IPC) in (Namik et al., 2011). A nonlinear pitch and torque controller using wind preview is designed in (Sarkar et al., 2020) and (Schlipf et al., 2013), giving promising results.

The novelty of this work is related to the FOWT damping analysis, i.e., the damping obtained by coupling the rotor and the platform pitch dynamics. This damping is explicitly derived from the coupled equations of rotor and floating platform. This analysis leads to the study of the instability phenomena underlining the conditions leading to the NMPZ. One new NMPZ, never discussed in literature, is discovered and analysed in this work. The domain of the instability of the platform is explicitly derived from the coupled system of equations. The control strategy proposed relies on this analysis and it allows to impose an explicit level of damping in the platform pitch motion to the controller without changing the period of platform pitching. This explicit form of the damping in the platform pitch dynamics is a novelty of this work.

The chosen strategy is, then, compared to one without platform pitch compensation (with detuning) and one that considers a single value of the compensation parameter for every wind speed and operating condition.

The document is organized as follows. In Section 2, the equations of the FOWT system are described with the considered degrees of freedom and their coupling terms. Section 2.1 presents the controller model considered in this work. The closed loop feedback system is then analysed in Section 2.3, leading to the definition of the conditions for the NMPZs. For this controller, a new control strategy dedicated to FOWTs, named $\zeta_{plt}$-fixed is presented and analytically derived in Section 2.4 and Section 2.5. Some numerical tests are presented and commented in Section 3.

## 2 Floating Offshore Wind Turbine and its controller

The floating offshore wind turbine (FOWT) is represented by a system of two degrees of freedom, namely the rotor speed $\Omega$ and the platform pitch angle, $\Phi$ as reported in Figure 1.

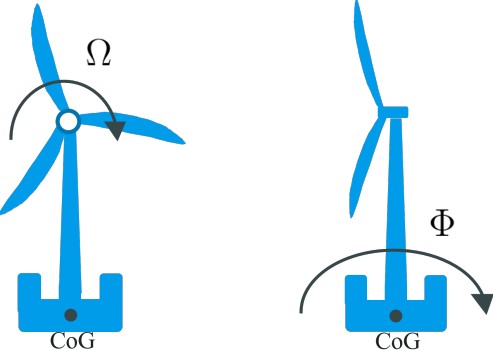

**Figure 1.** A scheme of the considered system with two degrees freedom, $\Omega$ around the rotor shaft and $\Phi$ around the center of gravity (CoG) of the system.

The surge degree of freedom is not considered in this model. In fact, the surge speed of the FOWT can be neglected with respect to the speed at nacelle generated by the pitch motion of the platform. This is already mentioned in (Sarkar et al., 2020b), where the authors remarked that the dynamics of the surge motion are much slower than those of the pitch. Hence, the surge can be considered as a static offset in the position of the wind turbine without any effects on the controller.

Two control parameters, $B$ (blade pitch) and $\tau_g$ (generator torque), and two external disturbances, $V$ (wind speed) and $W$ (wave elevation) are considered. For all the values that form a given operating point (namely $\Omega, \Phi, B, T_g, V, W$), the notation $X = \bar{x} + x$ is adopted, with $x$ being the small perturbation of a steady-state operating point $\bar{x}$.

The model is, then, based on the two fundamental equations:

$$\dot{\Omega}_g = \frac{N_g}{J_r}(T_a - N_g T_g) \tag{1}$$

$$J_t \ddot{\Phi} + D_t \dot{\Phi} + K_t \Phi = h_t F_a + \tau_{wave} \tag{2}$$

where $\Omega_g$ is the generator speed, hereafter noted $\Omega$, $T_a$ and $T_g$ are the aerodynamic and electric torque, $N_g$ is the gearbox ratio and $J_r$ is the rotor inertia. $J_t$ is the total system moment of inertia about the pitch rotation, $D_t$ is the natural damping coefficient (assumed constant), $K_t$ is a spring-like restoring coefficient (mainly given by the mooring lines of the floating platform), $h_t$ is the height of the rotor (approximately the tower length), $F_a$ is the aerodynamic force flowing from the rotor to the system and $T_{wave}$ is the overturning moment given by the waves.

Once a steady-state operating point $\bar{x}$ is reached, the same two equations can be applied to any small variations $x$ around this operating point. Equations (1) and (2) applied on $X$ can be written:

$$\dot{\omega} = \frac{N_g}{J_r}(\tau_a - N_g\tau_g) \tag{3}$$

$$J_t\ddot{\phi} + D_t\dot{\phi} + K_t\phi = h_t dF_a + \tau_{wave} \tag{4}$$

The infinitesimal thrust and torques satisfy (using the same notation $X = \bar{x} + x$): $T_g = \bar{\tau}_g + \tau_g$, $T_a = \bar{\tau}_a + \tau_a$, $T_{wave} = \bar{\tau}_w + \tau_{wave}$ and $F_a = \bar{F}_a + dF_a$ (notice that we do not use the notation $X = \bar{x} + x$ for $F_a$). By considering small perturbations of a steady-state operating point (given by $\Omega, \Phi, B, T_g, V, W$), it allows one to use the following linear forms:

$$\tau_a = \frac{\partial\tau_a}{\partial\omega}\omega + \frac{\partial\tau_a}{\partial v}v_r + \frac{\partial\tau_a}{\partial\beta}\beta \tag{5}$$

$$dF_a = \frac{\partial F_a}{\partial\omega}\omega + \frac{\partial F_a}{\partial v}v_r + \frac{\partial F_a}{\partial\beta}\beta \tag{6}$$

$$\tau_{wave} = \frac{\partial\tau_{wave}}{\partial w}w \tag{7}$$

The hypothesis of $\phi$ being small allows one to remove the terms $\frac{\partial\tau_a}{\partial\phi}\phi$, $\frac{\partial F_a}{\partial\phi}\phi$ and $\frac{\partial\tau_{wave}}{\partial\phi}\phi$.

The relative wind $v_r$ is the wind velocity in rotor reference frame, it is computed from $v$ by:

$$v_r = v - h_t\dot{\phi} \tag{8}$$

Under the assumption of $\phi$ being small, $h_t\dot{\phi}$ represents the rotor fore-aft velocity in a fixed global reference frame.

Equations (1) and (2) applied to small perturbations of a steady-state point can therefore be expressed in the linear form:

$$\dot{\omega} = \frac{N_g}{J_r}\left(\frac{\partial\tau_a}{\partial\omega}\omega + \frac{\partial\tau_a}{\partial v}(v - h_t\dot{\phi}) + \frac{\partial\tau_a}{\partial\beta}\beta - N_g\tau_g\right) \tag{1'}$$

$$J_t\ddot{\phi} + D_t\dot{\phi} + K_t\phi = h_t\left(\frac{\partial F_a}{\partial\omega}\omega + \frac{\partial F_a}{\partial v}(v - h_t\dot{\phi}) + \frac{\partial F_a}{\partial\beta}\beta\right) + \frac{\partial\tau_{wave}}{\partial w}w \tag{2'}$$

Those coupled second order equations yield the following four dimensional state-space model:

$$\dot{x} = A_0 x + B_c u_c + B_d u_d \tag{9}$$

Where $x = (\theta, \dot{\theta}, \phi, \dot{\phi})^T$, $\theta = \int\omega$ (i.e. $\dot{\theta} = \omega$), $u_c = (\beta, \tau_g)^T$ and $u_d = (v, w)^T$, and:

$$A_0 = \begin{bmatrix} 0 & 1 & 0 & 0 \\ 0 & \frac{N_g}{J_r}\frac{\partial\tau_a}{\partial\omega} & 0 & -h_t\frac{N_g}{J_r}\frac{\partial\tau_a}{\partial v} \\ 0 & 0 & 0 & 1 \\ 0 & \frac{h_t}{J_t}\frac{\partial F_a}{\partial\omega} & -\frac{K_t}{J_t} & -\frac{1}{J_t}\left(D_t + h_t^2\frac{\partial F_a}{\partial v}\right) \end{bmatrix} \quad B_c = \begin{bmatrix} 0 & 0 \\ \frac{N_g}{J_r}\frac{\partial\tau_a}{\partial\beta} & -\frac{N_g^2}{J_r} \\ 0 & 0 \\ \frac{h_t}{J_t}\frac{\partial F_a}{\partial\beta} & 0 \end{bmatrix} \quad B_d = \begin{bmatrix} 0 & 0 \\ \frac{N_g}{J_r}\frac{\partial\tau_a}{\partial v} & 0 \\ 0 & 0 \\ \frac{h_t}{J_t}\frac{\partial F_a}{\partial v} & \frac{1}{J_t}\frac{\partial\tau_{wave}}{\partial w} \end{bmatrix} \tag{10}$$

## 2.1 Control model description

The pitch controller model is described in this section.

The present control model considers $\omega_r$ as the reference for $\omega$ and $0$ as the reference for $\dot{\phi}$. It is based on several SISO (single-input-single-output) feedback loops. It can be seen as a multi-SISO:

- Proportional: $\beta_P = k_P(\Omega - \Omega_r)$

- Integral: $\beta_I = k_I \int (\Omega - \Omega_r)$

- Blade pitch ($\beta$) platform pitch compensation: $\beta_{comp} = k_\beta(\dot{\Phi}_r - \dot{\Phi})$

- Generator torque ($\tau_g$) platform pitch compensation: $\tau_{g,comp} = k_{\tau_g}(\dot{\Phi}_r - \dot{\Phi})$

Controllers described by the literature considering the same compensations (Abbas et al., 2022; Stockhouse et al., 2021) aim at maintaining $\omega$ steady near its rated value by acting on the blade pitch $\beta$ to vary the aerodynamic torque $\tau_a$ with the opposite sign with respect to the rotor infinitesimal speed $\omega = \Omega - \Omega_r$ , where the final goal is to obtain the same operational conditions of a bottom-fixed wind turbine. However, this strategy neglects the following phenomenon: the blade feather modifies the aerodynamic thrust $F_a$. Thus, a part of the opposing force on the platform is neglected. The strategy developed in this paper aims at minimizing $\phi$ variations with the constraint of maintaining a constant $\omega$. Such a control strategy should reduce the loads on the structures (nacelle, tower and floater). Section 3 considers a full aero-hydro-servo-elastic model to verify this assumption. The performance of the control strategy is analysed in a FOWT realistic environment reproduced by a numerical twin.

## 2.2 Global State-Space description

For a FOWT, the objective of the pitch control is to remain in the equilibrium operating point. It translates to: $\bar{\omega} = \Omega_r$ and $\bar{\dot{\phi}} = \dot{\Phi}_r = 0$. This objective allows one to justify the linear form of the global equations (1') and (2'). For constant inputs $\bar{v}$ and $\bar{w}$, this operating point is reached by the appropriated pitch ($\bar{\beta}$) and electric torque ($\bar{\tau}_g$).

The controller model is, here, introduced into the wind turbine state space description. For small perturbations of this steady-state operating point, the PI controller described previously becomes:

- Proportional:

$$\beta_P = k_P \omega \tag{11}$$

- Integral:

$$\beta_I = k_I \int \omega = k_I \theta \tag{12}$$

– Blade pitch ($\beta$) platform pitch compensation:

$$\beta_{comp} = -k_\beta \dot{\phi} \tag{13}$$

    – Generator torque ($\tau_g$) platform pitch compensation:

$$\tau_{g,comp} = -k_{\tau_g} \dot{\phi} \tag{14}$$

Figure 2 shows the entire picture of the controller model. This control strategy acts on the two dynamic systems, platform
and rotor. Hence, one can appreciate how the bottom-fixed scheme acting on the rotor speed ($\omega$) with a proportional integral
scheme is then corrected by the $\beta_{comp}$ (13) and the $\tau_g$ (14) that depend on the platform pitch speed error.

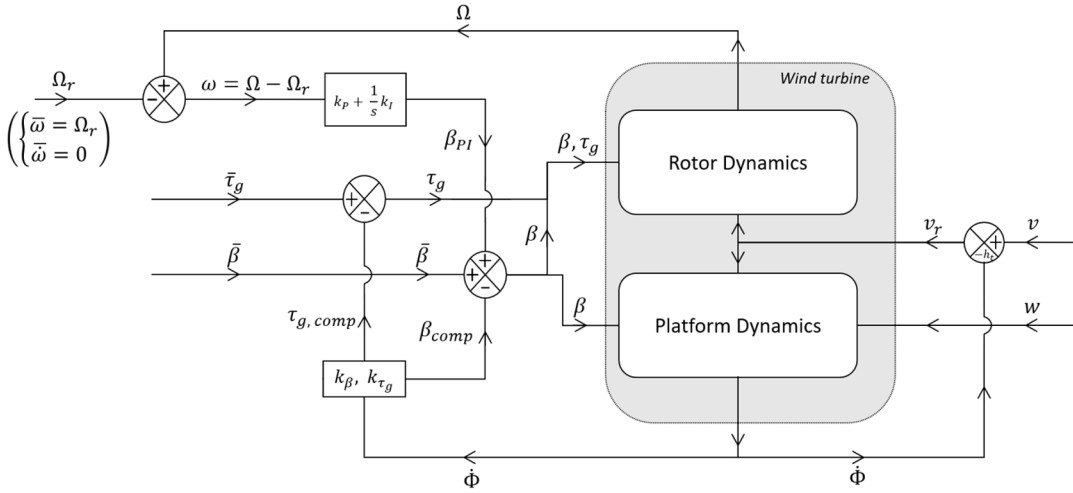

**Figure 2.** Block diagram of the controller model

The linear expression of $u_c = (\beta, \tau_g)^T$ as a function of $x = (\theta, \dot{\theta}, \phi, \dot{\phi})^T$ is $u_c = K_0 x + u_{c,ol}$ where:

$$K_0 = \begin{bmatrix} k_I & k_P & 0 & -k_\beta \\ 0 & 0 & 0 & -k_{\tau_g} \end{bmatrix} \tag{15}$$

is the matrix of the control gains and $u_{c,ol}$ is an optional additional control (open loop) that can be considered. This is useful
to analyse the NMPZ in the next section. By replacing it in eq. (9), it leads to:

$$\dot{x} = (A_0 + B_c K_0)x + B_c u_{c,ol} + B_d u_d \tag{16}$$

Which leads to define the global matrix of the closed loop system of equations:

$$A = (A_0 + B_c K_0) = \begin{bmatrix} 0 & 1 & 0 & 0 \\ k_I \frac{N_g}{J_r} \frac{\partial \tau_a}{\partial \beta} & \frac{N_g}{J_r}(\frac{\partial \tau_a}{\partial \omega} + k_P \frac{\partial \tau_a}{\partial \beta}) & 0 & \frac{N_g}{J_r}(-h_t \frac{\partial \tau_a}{\partial v} - k_\beta \frac{\partial \tau_a}{\partial \beta} + k_{\tau_g} N_g) \\ 0 & 0 & 0 & 1 \\ k_I \frac{h_t}{J_t} \frac{\partial F_a}{\partial \beta} & \frac{h_t}{J_t}(\frac{\partial F_a}{\partial \omega} + k_P \frac{\partial F_a}{\partial \beta}) & -\frac{K_t}{J_t} & \frac{-1}{J_t}(D_t + h_t^2 \frac{\partial F_a}{\partial v} + k_\beta h_t \frac{\partial F_a}{\partial \beta}) \end{bmatrix} \tag{17}$$

The time domain system can be rewritten in the Laplace complex domain. Using the following notation, $\mathscr{L}\{x(t)\} = \underline{X}$, eq. (16) translates into:

$$\underline{X} = (sI - A)^{-1}(B_c \underline{U}_{c,ol} + B_d \underline{U}_d). \tag{18}$$

By defining:

$$B = \begin{bmatrix} B_c & | & B_d \end{bmatrix}, \quad u = \begin{pmatrix} \beta_{ol} \\ \tau_{g,ol} \\ v \\ w \end{pmatrix} \tag{19}$$

and

$$G(s) = (sI - A)^{-1}B = \frac{1}{det(sI - A)}Com(sI - A)^T B = \frac{1}{\chi_A(s)}N(s), \tag{20}$$

it leads to:

$$\chi_A(s)\underline{X}(s) = N(s)\underline{U}(s) \tag{21}$$

G(s) is a $4 \times 4$ matrix. Every component of G(s) can be written as the quotient of a polynomial in $s$ and $\chi_a(s)$.

## 2.3 Non minimum phase zeros analysis and resolution (negative damping on the control)

This section analyzes the problem of negative damping by addressing the positions of the zeros of each component of $G$ in the complex plane, i.e. the points where eq. (21) is well defined but becomes:

$$\chi_A X_0^T \cdot \underline{X} = 0. \tag{22}$$

This translates the fact that $s$ is a NMPZ if there exists a specific $X_0^T$, such as, for any value of $\underline{U}(s)$, the linear combination $X_0^T N(s)\underline{U}(s)$ gives $X_0^T = 0$. Here, $s$ is formally the complex Laplace variable, so that formally, $s \in \{x + iy, \ x > 0\}$. Even though eq. (22) can be defined on the whole complex plane, only zeros with a strictly positive real part are NMPZs. The reader is referred to (Hoagg and Bernstein, 2007) for a complete description of NMPZs.

Physically a $X_0^T$ is equivalent to an infinitesimal shifting along a specific direction of a steady-state point that can not be obtained with any infinitesimal shifting of the input. This phenomenon is better illustrated case by case.

For the rest of the section, an open loop control on $\beta$ is considered in order to highlight the NPMZ. Hence, $\beta_{ol}$ is added to the multiple SISO as already described in eq. (16). Since the feedback control on $\beta$ can not erase the NMPZ condition, to lighten the formulas, the notation $\beta$ will be used instead of $\beta_{ol}$ in this section.

### 2.3.1  $\phi$-NMPZ: negative damping on $\phi$ control by $\beta$

The gain equation in the Laplace domain for $\beta \to \phi$ control is obtained by projecting eq. (21) on the $x = (0, 0, \phi, 0)$ axis and considering only a $\beta$ perturbation, i.e. an input $u = (\beta, 0, 0, 0)$. The resulting equation is:

$$\chi_A(s)\underline{\phi}(s) = N_{3,1}(s)\underline{\beta}(s), \tag{23}$$

$$N_{3,1}(s) = \frac{J_r}{N_g} \frac{\partial F_a}{\partial \beta} s^2 + \left( \frac{\partial \tau_a}{\partial \beta} \frac{\partial F_a}{\partial \omega} - \frac{\partial F_a}{\partial \beta} \frac{\partial \tau_a}{\partial \omega} \right) s \tag{24}$$

The condition for the NMPZ on $\beta \to \phi$ control is that $N_{3,1}$ has a root with a real part strictly positive. Assuming that $\beta = \beta_f$, the fine pitch, the previous derivatives are all negative. Hence, the root research of $N_{3,1}$ leads to:

$$\frac{\partial \tau_a}{\partial \omega} \Big/ \frac{\partial \tau_a}{\partial \beta} \quad < \quad \frac{\partial F_a}{\partial \omega} \Big/ \frac{\partial F_a}{\partial \beta} \tag{25}$$

Intuitively, this corresponds to an operating point where $\tau_a$ *is rather influenced by* $\beta$ and $F_a$ *is rather influenced by* $\omega$. This NMPZ does not depend on parameters of the platform, it is only related to Wind Turbine Generator (WTG) performances. However, the importance of the phenomenon is related to the platform properties. It is to be noted that this NMPZ has never been highlighted in literature and the controller model (with compensations of the platform motions) introduced in Section 2.1 in the feedback control loop does not prevent it. It results only from the characteristic of the FOWT system. Further works should focus on this phenomenon and introduce corrections to prevent it for any FOWT system.

In absence of NMPZ, i.e. eq. (25) being false, increasing $\beta$ from a steady-state operating point (i.e. setting $d\ddot{\beta} > 0$, $d\dot{\beta} > 0$ and $d\beta > 0$) will always imply a reduction of $\phi$. In presence of NMPZ, i.e. if eq. (25) is true, the reduction or the increase of $\phi$ (with respect to the operating point) depends on the ratio between $\ddot{\beta}$ and $\dot{\beta}$.

When eq. (25) is verified, it means that: $\tau_a$ is more sensitive to blade pitch ($\beta$) than rotational speed ($\omega$) and $F_a$ is more sensitive to $\omega$ than $\beta$. Therefore, by increasing $\beta$, $\omega$ increases and then, it occasions $F_a$ to decrease. Then, $\phi$ increases. If eq. (25) is not verified, increasing blade pitch $\beta$ from a steady-state operating point always reduces platform pitch $\phi$. In practice, this effect can become an issue for a control algorithm mainly focused on $\omega$ stabilization since it generates unexpected platform dynamics. For a more detailed approach of the initial undershoot phenomenon, the reader is referred to (Hoagg and Bernstein, 2007).

Figure 3 reproduces in the time domain $\phi$ and $\omega$ responses to a $\beta$-step input (at $t = 10s$): values (resumed in Table 1) are chosen arbitrarily. so that eq. (25) is false: $\phi$ decreases. On the right, values are chosen so that the eq. (25) is true: $\phi$ increases

even though $\beta$ has step up. The chose values are not intended to simulate a real turbine, but only illustrate the described phenomena. Section 3 focuses on more realistic FOWT tests.

**Table 1.** The set of parameters to show the NMPZ of eq. (25). They are not intended to simulate a real turbine.

| | eq. (25) false | eq. (25) true |
|---|---|---|
| $\frac{\partial \tau_a}{\partial v}$ | $2980.9 \ kN.s$ | $3079 \ kN.s$ |
| $\frac{\partial F_a}{\partial v}$ | $354.8 \ kN.s.m^{-1}$ | $355.6 \ kN.s.m^{-1}$ |
| $\frac{\partial \tau_a}{\partial \omega}$ | $-58597.1 \ kN.m.s.rad^{-1}$ | $-55499.5 \ kN.m.s.rad^{-1}$ |
| $\frac{\partial F_a}{\partial \omega}$ | $-5658.0 \ kN.s.rad^{-1}$ | $-5820.4 \ kN.s.rad^{-1}$ |
| $\frac{\partial \tau_a}{\partial \beta}$ | $-152347.8 \ kN.m.rad^{-1}$ | $-160140.5 \ kN.m.rad^{-1}$ |
| $\frac{\partial F_a}{\partial \beta}$ | $-16052.2 \ kN.rad^{-1}$ | $-15260 \ kN.rad^{-1}$ |

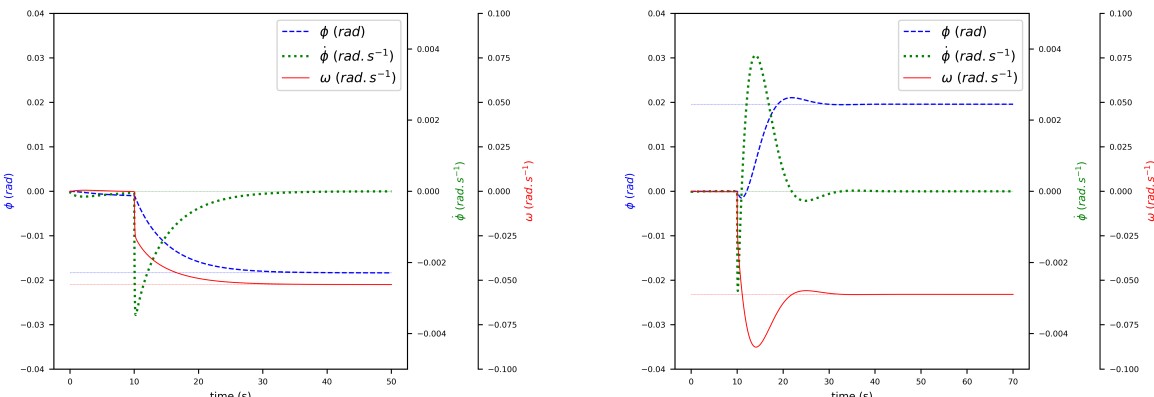

**Figure 3.** Platform pitch $\phi$ and rotor speed $\omega$ responses to a blade pitch $\beta$-step input (at $t = 10s$): on the left, values (Table 1) are chosen so that eq. (25) is false: $\phi$ decreases. On the right, values (Table 1) are chosen so that eq. (25) is true: $\phi$ increases even though $\beta$ has step up.

### 2.3.2 $\omega$-NMPZ: negative damping on $\omega$ control by $\beta$

Gain equation in the Laplace domain for $\beta \to \omega$ control is given by:

$$\chi_A(s) \, \underline{\omega}(s) = N_{2,1}(s) \, \underline{\beta}(s) \tag{26}$$

$$N_{2,1}(s) = \frac{J_t}{h_t} \frac{\partial \tau_a}{\partial \beta} s^3 + \left[ \frac{D_t}{h_t} \frac{\partial \tau_a}{\partial \beta} + h_t \left( \frac{\partial \tau_a}{\partial \beta} \frac{\partial F_a}{\partial v} - \frac{\partial F_a}{\partial \beta} \frac{\partial \tau_a}{\partial v} \right) + k_{\tau_g} N_g \frac{\partial F_a}{\partial \beta} \right] s^2 + \frac{K_t}{h_t} \frac{\partial \tau_a}{\partial \beta} s \tag{27}$$

Hence the condition for NMPZ on $\beta \to \omega$ control is:

$$h_t^2 \left( \frac{\partial F_a}{\partial v} - \left( \frac{\partial \tau_a}{\partial v} - k_{\tau_g} \frac{N_g}{h_t} \right) \frac{\frac{\partial F_a}{\partial \beta}}{\frac{\partial \tau_a}{\partial \beta}} \right) < -D_t \tag{28}$$

This corresponds with an operating point where $\tau_a$ *is rather influenced by* $v$ and $F_a$ *is rather influenced by* $\beta$. In presence of NMPZ, i.e. if eq. (28) is true, the sign of $d\omega$ depends on the choice of $d\ddot{\beta}$, $d\dot{\beta}$ and $d\beta$. Intuitively, the latter only happens when eq. (28) is verified: increasing blade pitch will *reduce $F_a$ more than it increases $\tau_a$* (because $F_a$ *is rather influenced by* $\beta$), thus $\dot{\phi}$ will decrease and cause relative wind $v_r = v - h_t\dot{\phi}$ to increase. As $\tau_a$ *is rather influenced by* $v$, this will reduce $\omega$ in the end.

In practice, this effect can become an issue if a $\omega$ control algorithm obtains the opposite result than what was expected.

$$k_{\tau_g} = m_{\tau_g} \frac{h_t}{N_g} \frac{\partial \tau_a}{\partial v}, \quad m_{\tau_g} \in [0,1] \tag{29}$$

In absence of NMPZ, i.e. eq. (28) being false, increasing $\beta$ from a steady-state operating point will always imply reducing $\omega$. In order to visualize this NMPZ, Figure 4 shows $\omega$ responses to a $\beta$-step input (at $t = 10s$). On the left, parameters (Table 2) are chosen so that the condition eq. (28) is false: $\omega$ decreases. On the right, parameters are chosen so that condition eq. (28)

is true: at first $\omega$ increases even though $\beta$ has step up.

**Table 2.** The set of parameters chosen to show the NMPZ of eq. (28). They are not intended to simulate a real turbine.

| | eq. (28) false | eq. (28) true |
|---|---|---|
| $\frac{\partial \tau_a}{\partial v}$ | $2980.9 \; kN.s$ | $2838 \; kN.s$ |
| $\frac{\partial F_a}{\partial v}$ | $354.8 \; kN.s.m^{-1}$ | $303.0 \; kN.s.m^{-1}$ |
| $\frac{\partial \tau_a}{\partial \omega}$ | $-58597.1 \; kN.m.s.rad^{-1}$ | $-59428.7 \; kN.m.s.rad^{-1}$ |
| $\frac{\partial F_a}{\partial \omega}$ | $-5658.0 \; kN.s.rad^{-1}$ | $-6282.9 \; kN.s.rad^{-1}$ |
| $\frac{\partial \tau_a}{\partial \beta}$ | $-152347.8 \; kN.m.rad^{-1}$ | $-133058.7 \; kN.m.rad^{-1}$ |
| $\frac{\partial F_a}{\partial \beta}$ | $-16052.2 \; kN.rad^{-1}$ | $-18247.0 \; kN.rad^{-1}$ |

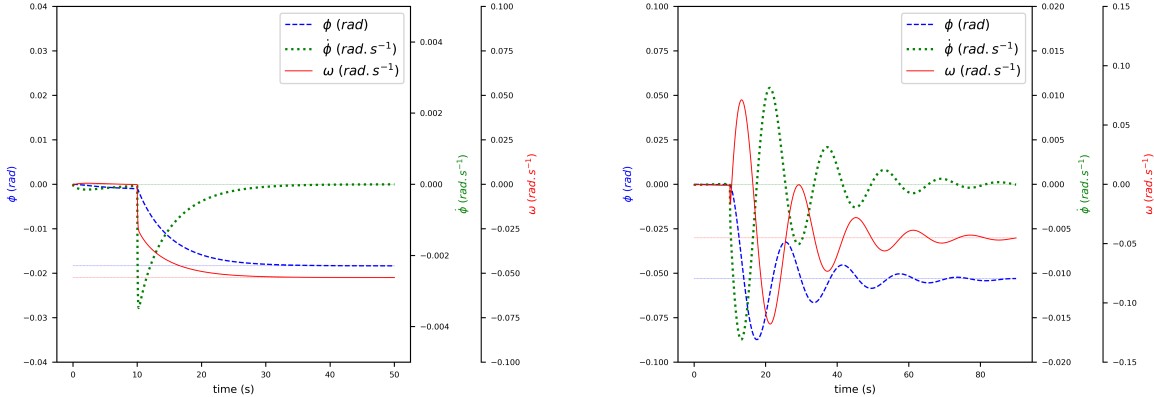

**Figure 4.** Rotor speed ($\omega$) responses to a blade pitch ($\beta$)-step input (at $t = 10s$). On the left, parameters (Table 2) are chosen so that the condition eq. (28) is false: $\omega$ decreases. On the right, parameters are chosen so that condition eq. (28) is true: at first $\omega$ increases even though $\beta$ has step up.

### 2.3.3 NMPZs and stability analysis

Comparison between Figures 3 and 4 enlightens what really happens after a step input, with and without NMPZ: at the beginning both $\omega$ and $\dot{\phi}$ always decrease just after the step. However, when both NMPZ conditions eq. (25) and eq. (28) are false, those tendencies do not change. Conversely, when eq. (25) is true, we observe that $|\dot{\omega}|$ is so big that $\dot{\phi}$ jumps into positive values. Similarly, when eq. (28) is true, we observe that $|\dot{\phi}|$ is so big that $\omega$ jumps (only for a short time) into positive values. NMPZ, as we have seen in the examples, can cause important shifts and unexpected behaviors for both $\omega$ and $\phi$.

The NMPZ $\beta \rightarrow \phi$ does not depend on above defined parameters. Consequently, the model in Section 2.1 does not prevent it . However condition eq. (25) forecasts which operating points it affects. On the other hand, a wise choice of $\tau_g$ avoids $\beta \rightarrow \omega$ NMPZ, which is the main reason why this compensation has already been introduced by Stockhouse et al. (2021) and in the controller model introduced in Section 2.1.

In order to complete the analysis of NMPZ phenomena related to FOWT system, a hypothetical situation where both conditions eq. (25) and eq. (28) are true has been simulated and reported in Figure 5. At first, the dynamics are always the same: both $\dot{\phi}$ and $\omega$ decrease, but soon they both diverge because of the NMPZ phenomena (combined with the closed loop control). The Pole-Zero plots in Figure 6 will lead to a better understanding of this instability. It is to be noted that, $k_P/k_I$ corrections (without compensations $\tau_g$ and $k_\beta$) can delay this divergence but can not avoid it.

**Table 3.** The set of parameters chosen to show the instability given by NMPZs of eq. (25) and eq. (28) . They are not intended to simulate a real turbine.

| | eq. (25) and eq. (28) true |
|---|---|
| $\frac{\partial \tau_a}{\partial v}$ | $3105.0 \, kN.s$ |
| $\frac{\partial F_a}{\partial v}$ | $293.0 \, kN.s.m^{-1}$ |
| $\frac{\partial \tau_a}{\partial \omega}$ | $-51356.5 \, kN.m.s.rad^{-1}$ |
| $\frac{\partial F_a}{\partial \omega}$ | $-7150.0 \, kN.s.rad^{-1}$ |
| $\frac{\partial \tau_a}{\partial \beta}$ | $-148063.0 \, kN.m.rad^{-1}$ |
| $\frac{\partial F_a}{\partial \beta}$ | $-16543.6 \, kN.rad^{-1}$ |

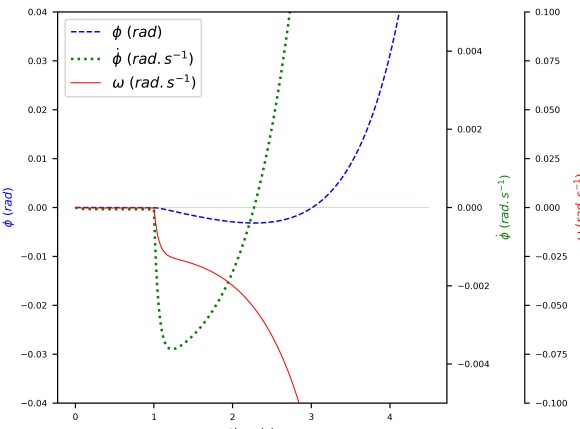

**Figure 5.** an hypothetical situation where both conditions eq. (25) and eq. (28) are true. At first, both platform pitch rotational velocity $\dot{\phi}$ and rotor speed $\omega$ decrease, but soon they both diverge because of the negative damping ($k_P/k_I$ corrections can delay this divergence but can not avoid it).

A Pole-Zero plot is a commonly used synthesis of both NMPZ and stability issues. The above case by case analysis high-lighted the drawback of allowing a zero of the transfer function in the right half of the complex plane. Similarly, the stability of a system can be well synthesized by the position of the poles of the transfer function. Poles of the transfer function situated in the right half of the plane result in a global instability such as the one observed in Figure 5.

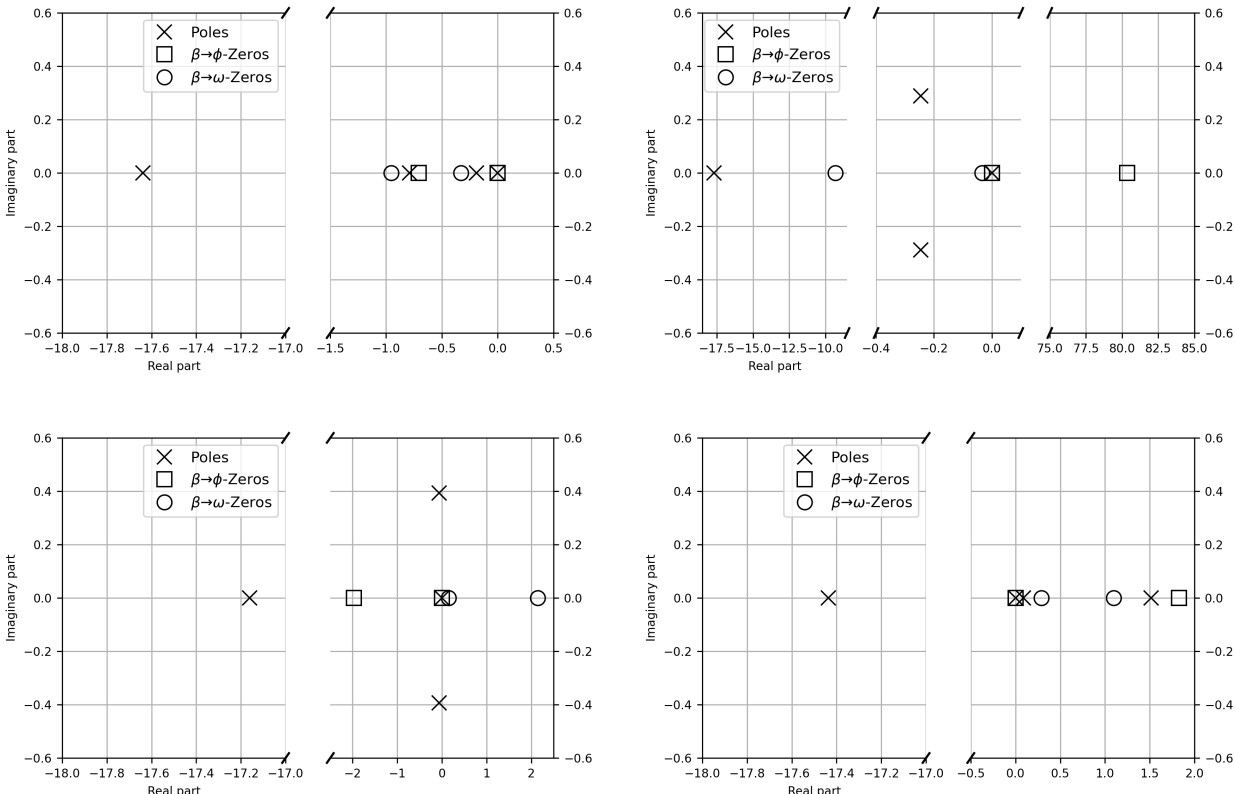

**Figure 6.** Pole-Zero analysis of the systems described in tables 1, 2, and 3: the upper-left plot corresponds to an hypothetical situation where both equations (25) and (28). are false, the upper-right to a situation where only eq. (25) is true, the lower-left to a situation where only eq. (28) is true and the lower-right to a situation where both equations (25) and (28) are true.

Figure 6 shows, for every hypothetical situation described above (see tables 1, 2, and 3), the position of poles and zeros of the transfer functions $G_{3,1}$, describing $\phi$ control by $\beta$ (see Section 2.3.1) and $G_{2,1}$ describing $\omega$ control by $\beta$ (see Section 2.3.2). $G_{3,1}$ and $G_{2,1}$ have the same denominator, which is the complex polynomial $\chi_A$, see eq. (20). Thus, they have the same poles. Their numerators are respectively $N_{3,1}$ and $N_{2,1}$. In the upper-left plot, both equations (25) and (28) are false. There are no NMPZ: indeed, all the zeros are in the left half of the complex plane. In the upper-right plot, only eq. (25) is true. All the zeros

of $N_{2,1}$ are in the left half of the complex plane while one zero of $N_{3,1}$ is in its right half: there is one NMPZ corresponding to the $\phi$ control by $\beta$. Similarly, in the lower-left plot, only eq. (28) is true. Only two zeros of $N_{2,1}$ are in its right half of the complex plane: those are NMPZs corresponding to the $\omega$ control by $\beta$. Finally, in the lower-right plot both equations (25) and (28) are true: there are zeros of $N_{3,1}$ and of $N_{2,1}$ in the right half of the complex plane. Moreover, in this case, we also find two poles of the transfer function in the right half of the complex plane: this is consistent with the time evolution plotted in Figure

5, where one can observe an instability.

### 2.3.4 compensation of the ω-NMPZ

The issue related to the $\omega$-NMPZ has been presented in Section 2.3.2. Literature has addressed this NMPZ phenomenon and suggested several control corrections. Due to the nature of this phenomenon, any correction concerning $\beta$ control, introduced in eq. (13) as $d\beta_{comp} = -k_\beta\dot{\phi}$, cannot prevent completely this NMPZ. However, detuning the PI controller (by lowering $k_P$ and $k_I$ gains) or using the $\beta$ platform pitch compensation as suggested in (Abbas et al., 2022), can mitigate the effect of NMPZ when eq. (28) is true.

The complete prevention of the problem can be obtained by several sets of parameters that involve the WTG, the floating platform and the control set-up. In fact, for this NMPZ, eq. (14) of the controller model described in Section 2.1 allows one to avoid the NMPZ by choosing a well-suited value of $k_{\tau_g}$. This compensation has been already introduced by Fischer (2013) and Stockhouse et al. (2021) with the formula:

$$k_{\tau_g} = m_{\tau_g}\frac{h_t}{N_g}\frac{\partial\tau_a}{\partial v}, \quad m_{\tau_g} \in [0,1] \tag{30}$$

It is to be noted that, usually, it needs to be saturated because of turbine generator design constraints concerning the generator torque. Pole-Zero plots are useful to get a better understanding about the choice of the parameter $k_{\tau_g}$, and more precisely the effect of varying the coefficient $m_{\tau_g}$.

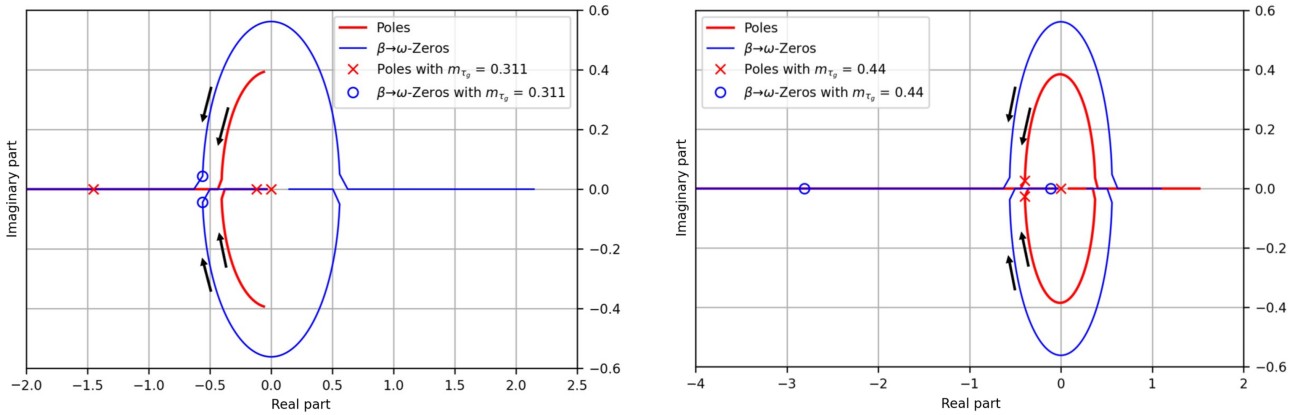

**Figure 7.** Pole-Zero analysis of the systems described in tables 1 and 3, with the parameter $m_{\tau_g}$ varying from 0 to 1. According to the value of $m_{\tau_g}$, zeros and poles of the system move from the right to the left half of the complex domain. The left plot corresponds to an hypothetical situation where, supposing $m_{\tau_g} = 0$, eq. (25) would be true and eq. (28) would be false (Pole-Zero plot reported in the upper-right of Figure 6). The right plot corresponds to a situation where supposing $m_{\tau_g} = 0$, both equations (25) and (28) would be true (Pole-Zero plot reported in the lower-right of Figure 6).

In the left plot of Figure 7, positions of poles and zeros of the transfer function vary with $m_{\tau_g}$. Both zeros, initially (i.e. with $m_{\tau_g} = 0$) in the right half of the complex plane, end up in its left half: the NMPZ issue is solved. One should also notice

that the poles are also displaced but remain in the left half of the complex plane. The stability margin, however, might change. Thus, while choosing an explicit value of $m_{\tau_g}$, one should take into account not only the position of the zeros but also the poles of the transfer function. Here, the stability margin is not maximized by the value chosen as example, $m_{\tau_g} = 0.311$.

The right plot of Figure 7 corresponds to an instability situation: initially (i.e. with $m_{\tau_g} = 0$) not only both zeros are in the right half of the complex plane, also two poles are there. Again, both the zeros and the poles are displaced as $m_{\tau_g}$ varies from 0 to 1. For a wisely chosen value of $m_{\tau_g}$ (here $m_{\tau_g} = 0.44$) they both end up in the left half of the complex plane and the stability margin can be maximized. In general, the choice of a value for $m_{\tau_g}$ should also take into account a possible saturation due to turbine generator design constraints.

## 295   2.4   Damping analysis

In Section 2.3 the issue of NMPZ, i.e. the issue of negative damping in the *control/input side* of the equation, is analysed. The influence of the gains $k_P$, $k_I$, $k_\beta$ and $k_{\tau_g}$ on the damping of the system (cf. Section 2.3) is investigated within the analytical framework set in the previous sections. The goal is to optimize (or tune) the stability of $\omega$ and $\phi$ responses to an external ($v$ and $w$) disturbance. In other words, the goal is to obtain an explicit expression of the damping of the FOWT system with respect to

the control parameters, $k_P$, $k_I$, $k_\beta$ and $k_{\tau_g}$, such that, for an imposed level of damping, one can obtain a value of the control parameters. This is a powerful result for the floating wind community and a novelty of this work with respect to the existing literature.

Considering the whole system, with both degrees of freedom $\omega$ and $\phi$ and their coupling, in the complex domain, leads to the eq. (21). The study of the damping is related to $\chi_A(s) = det(sI - A)$, defined in eq. (20). The explicit form of $\chi_A$ is:

$$\chi_A(s) = \chi_{rot}(s)\chi_{plt}(s) + \frac{N_g h_t}{J_r J_t} s \left[ (k_p s + k_I)\frac{\partial F_a}{\partial \beta} h_t \frac{\partial \tau_a}{\partial v} - \left( \frac{J_r}{N_g}\frac{\partial F_a}{\partial \beta}(k_P s + k_I) + \frac{\partial F_a}{\partial \omega} \right)\left( k_\beta \frac{\partial \tau_a}{\partial \beta} + k_{\tau_g} N_g \right) \right] \tag{31}$$

where:

$$\chi_{rot}(s) = s^2 - \frac{N_g}{J_r}\frac{\partial \tau_a}{\partial \omega} s - \frac{N_g}{J_r}\frac{\partial \tau_a}{\partial \beta}(k_P s + k_I)$$
$$\chi_{plt}(s) = s^2 + \frac{1}{J_t}\left(D_t + h_t^2\frac{\partial F_a}{\partial v} + k_\beta h_t \frac{\partial F_a}{\partial \beta}\right)s + \frac{K_t}{J_t} \tag{32}$$

The term in square parenthesis represents the coupling term between the dynamics of the platform ($\phi$) and the dynamics of the rotor ($\omega$).

In this coupled form, it is complicated to explicitly determine the damping of the system. In the next paragraph, under some hypothesis, the coupled system can be separated in two second order systems, one related to the rotor dynamics $\omega$ and the other one related to the floating dynamics $\phi$. In particular, for the latter, it is possible to define a damping for the floating platform and obtain an explicit form for the compensation term $k_\beta$ related to the imposed damping.

### 2.4.1 Simplified analysis of rotor dynamics:

Defining a damping coefficient (or a damping ratio) requires to reduce the global system to a second order oscillatory system. Equations (16) and (17) couple rotor and platform pitch dynamics, they hence involve a $4^{th}$ order polynomial expression. In order to deal with rotor dynamics independently of the platform, it is supposed:

$$h_t \dot{\phi} \ll v \tag{33}$$

For large FOWT systems, this hypothesis is, generally, respected. It implies:

$$N_g k_{\tau_g} \dot{\phi} \ll \frac{\partial \tau_a}{\partial v} v$$
$$\frac{\partial \tau_a}{\partial \beta} k_\beta \dot{\phi} \ll \frac{\partial \tau_a}{\partial v} v \tag{34}$$

Under such assumptions, the linear form of eq. (1) becomes:

$$\dot{\omega} = \frac{N_g}{J_r} \left( \frac{\partial \tau_a}{\partial \omega} \omega + \frac{\partial \tau_a}{\partial v} v + \frac{\partial \tau_a}{\partial \beta} \beta - \tau_g \right) \tag{1"}$$

and the control is described by the PI controller: $\dot{\beta} = k_P \dot{\omega} + k_I \omega$, so that the resulting Laplace transform equation is

$$\underline{\omega}(s) = G_{rot}(s) \underline{v}(s) \tag{35}$$

where, considering a $k_I > 0$,

$$G_{rot}(s) = \frac{\frac{\partial \tau_a}{\partial v} s}{s^2 - \frac{N_g}{J_r} \frac{\partial \tau_a}{\partial \omega} s - \frac{N_g}{J_r} \frac{\partial \tau_a}{\partial \beta} (k_P s + k_I)},$$

$$i.e. \quad G_{rot}(j\nu) = \frac{1}{1 + \frac{j}{2\zeta_{rot}} \left( \frac{\nu}{\nu_{rot}} - \frac{\nu_{rot}}{\nu} \right)} \frac{-\frac{\partial \tau_a}{\partial v}}{\frac{N_g}{J_r} \frac{\partial \tau_a}{\partial \omega} + \frac{N_g}{J_r} \frac{\partial \tau_a}{\partial \beta} k_P} \tag{36}$$

with:

$$\nu_{rot} = \sqrt{-\frac{N_g}{J_r} \frac{\partial \tau_a}{\partial \beta} k_I}, \quad \zeta_{rot} = -\frac{\frac{N_g}{J_r} \frac{\partial \tau_a}{\partial \omega} + \frac{N_g}{J_r} \frac{\partial \tau_a}{\partial \beta} k_P}{2\sqrt{-\frac{N_g}{J_r} \frac{\partial \tau_a}{\partial \beta} k_I}} \tag{37}$$

Thus, when all interactions with platform pitch are neglected, the rotor behaves like a second order oscillatory system. The corresponding filter $G_{rot}$ is a second order band-pass filter with cutoff angular frequency $\nu_{rot}$ [1].

The above formulas enable one to obtain explicitly $k_I$ and $k_P$.

They are well known: several controllers, such as (Abbas et al., 2022), suggest to define:

$$|k_I| = \left| \frac{\nu_{rot}^2}{\frac{N_g}{J_r} \frac{\partial \tau_a}{\partial \beta}} \right| \quad \text{and} \quad |k_P| = \left| \frac{\left( \frac{N_g}{J_r} \frac{\partial \tau_a}{\partial \omega} + 2\zeta_{rot}\nu_{rot} \right)}{\frac{N_g}{J_r} \frac{\partial \tau_a}{\partial \beta}} \right| \tag{38}$$

[1]In case $k_I \leq 0$, $\nu_{rot}$ and $\zeta_{rot}$ would be imaginary according to the formulas above. $G_{rot}$ would be no longer a band-pass filter.

### 2.4.2 Simplified analysis of platform dynamics:

Similarly to what is done in the previous paragraph, here the global system of equations (1, 2) is reduced to a second order oscillatory system that allows us to have a better understanding of platform dynamics.

Considering $k_P = k_I = 0$ and assuming:

$$\frac{\partial F_a}{\partial \omega}\omega << \frac{\partial F_a}{\partial v}v_r + \frac{\partial F_a}{\partial \beta}\beta \tag{39}$$

The latter is the condition to decouple the global system. It enables to consider $\phi$ response independently of $\omega$, and as a second

order oscillatory system's degree of freedom. The resulting Laplace transform equation is:

$$\begin{pmatrix} \underline{\phi} \\ \underline{\dot{\phi}} \end{pmatrix}(s) = G_{plt}(s)\underline{u}_d(s) \tag{40}$$

$u_d = \begin{pmatrix} v \\ w \end{pmatrix}$ : the input array is reduced because only the damping in the *output side* is analysed and it is not necessary for this to consider any additional control input.

$$G_{plt}(s) = (sI - A_{plt})^{-1}B_d \tag{41}$$

$$A_{plt} = \begin{bmatrix} 0 & 1 \\ -\frac{K_t}{J_t} & \frac{-1}{J_t}(D_t + h_t^2 \frac{\partial F_a}{\partial v} + k_\beta h_t \frac{\partial F_a}{\partial \beta}) \end{bmatrix} \tag{42}$$

$A_{plt}$ is the bottom-right part of $A$ defined in eq. (17).

Looking at the $\phi$ degree of freedom, eq. (40) gives:

$$\underline{\phi}(s) = G_{plt,1,1}(s)\underline{v}(s) + G_{plt,1,2}(s)\underline{w}(s), \tag{43}$$

with:

$$(G_{plt,1,1}, G_{plt,1,2})(s) = \left( \frac{\frac{h_t}{J_t}\frac{\partial F_a}{\partial v}}{s^2 + \frac{1}{J_t}(D_t + h_t^2 \frac{\partial F_a}{\partial v} + k_\beta h_t \frac{\partial F_a}{\partial \beta})s + \frac{K_t}{J_t}}, \frac{\frac{1}{J_t}\frac{\partial \tau_{wave}}{\partial w}}{s^2 + \frac{1}{J_t}(D_t + h_t^2 \frac{\partial F_a}{\partial v} + k_\beta h_t \frac{\partial F_a}{\partial \beta})s + \frac{K_t}{J_t}} \right), \tag{44}$$

i.e.:

$$(G_{plt,1,1}, G_{plt,1,2})(j\nu) = \frac{1}{1 - \left(\frac{\nu}{\nu_{plt}}\right)^2 + 2j\zeta_{plt}\frac{\nu}{\nu_{plt}}} \left( \frac{h_t}{K_t}\frac{\partial F_a}{\partial v}, \frac{1}{K_t}\frac{\partial \tau_{wave}}{\partial w} \right), \tag{45}$$

$$\nu_{plt} = \sqrt{\frac{K_t}{J_t}}, \qquad \zeta_{plt} = \frac{1}{2\sqrt{K_t J_t}}\left( D_t + h_t^2 \frac{\partial F_a}{\partial v} + k_\beta h_t \frac{\partial F_a}{\partial \beta} \right) \tag{46}$$

Thus, when all interactions with rotor dynamics are neglected, the platform behaves like a second order oscillatory system. The corresponding filter $G_{plt}$ is a second order low-pass filter with cutoff angular frequency defined by $\nu_{plt}$ and damping ratio defined by $\zeta_{plt}$.

## 2.5 Artificial damping of the platform: $\zeta_{plt}$-fixed strategy

By knowing the features of the FOWT, one can impose a given level of damping and obtain an explicit expression for the $k_\beta$:

$$k_\beta = \frac{1}{h_t \frac{\partial F_a}{\partial \beta}} \left( 2\sqrt{K_t J_t} \zeta_{plt} - D_t - h_t^2 \frac{\partial F_a}{\partial v} \right) \tag{47}$$

The strategy is such that $k_\beta$ is a negative number instead of what is proposed in the literature [2]. In (Stockhouse et al., 2021), $\beta_{comp} = -k_\beta \dot{\phi}$ is introduced in order to erase at first order the coupling between platform and rotor dynamics, and therefore $k_\beta$ is positive, defined by:

$$k_\beta = -h_t \frac{\partial \tau_a}{\partial v} \Big/ \frac{\partial \tau_a}{\partial \beta} \tag{48}$$

In Lackner et al. (2009), a platform pitch control involving a parameter equivalent to $k_\beta$ is assessed, and a numerical approach leads to defining a $k_\beta < 0$, as for eq. (47), but unique. Here, eq. (47) provides the reader with an explicit value of $k_\beta$ that is different for any turbine and floater characteristics and operating point.

### 2.5.1 Expected effect on platform dynamics

Figure 8 shows Bode diagram of the second order low-pass filter $G_{plt}$. In other words, it shows how $\zeta_{plt}$ value can affect the damping of platform oscillations.

It can be observed that $\zeta_{plt}$ has a significant effect on the damping of platform oscillations only for angular frequencies $\nu \approx \nu_{plt,natural}$. The yellow vertical band in Figure 8 shows the interval of angular frequencies $I_{damped}$, arbitrarily defined by:

$$I_{damped} = \left[ \frac{\nu_{plt,natural}}{\sqrt{2}} \ , \ \sqrt{2}\nu_{plt,natural} \right] \tag{49}$$

that are directly damped when $\zeta_{plt}$ increases. Therefore, it is to be expected that $\zeta_{plt}$-fixed strategy will be well fit to reduce platform motion and tower loads when their variations happen at an angular frequency $\nu \in I_{damped}$.

---

[2]In (Abbas et al., 2022), $\beta_{comp}$ is defined as in (Stockhouse et al., 2021) but with the convention $\beta_{comp} = k_{float}\dot{\phi}$ so that $k_{float} = -k_\beta = h_t \frac{\partial \tau_a}{\partial v} \Big/ \frac{\partial \tau_a}{\partial \beta}$ is negative. It would be positive with the convention used in this work and in (Stockhouse et al., 2021).

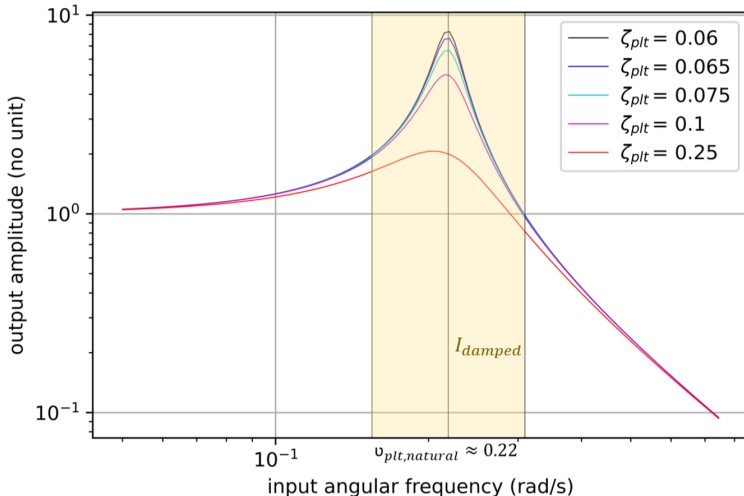

**Figure 8.** Bode diagram of second order low-pass filter $G_{plt}$

### 2.5.2 Expected effect on rotor dynamics

In this part the first order effect of $\zeta_{plt}$-fixed strategy on the rotor dynamics is analysed. The state space representation of the FOWT dynamics is given by eq. (16). By considering a disturbance and open-loop input equal to $0$ (i.e. $u_{c,ol} = u_d = 0$), this
leads to the following linear equation, truncated at first order:

$$\dot{\omega} = \ddot{\theta} = A_{2,1}\theta + A_{2,2}\dot{\theta} + A_{2,4}\dot{\phi}, \tag{50}$$

where

$$
\begin{aligned}
A_{2,4} &= \frac{N_g}{J_r}\left(-h_t\frac{\partial \tau_a}{\partial v} - k_\beta\frac{\partial \tau_a}{\partial \beta} + k_{\tau_g}N_g\right) \\
&= \frac{N_g}{J_r}\left(-h_t\frac{\partial \tau_a}{\partial v} + \frac{\frac{\partial \tau_a}{\partial \beta}}{h_t\frac{\partial F_a}{\partial \beta}}\left(D_t + h_t^2\frac{\partial F_a}{\partial v} - 2\sqrt{K_t J_t}\zeta_{plt}\right) + k_{\tau_g}N_g\right).
\end{aligned}
\tag{51}
$$

Moreover, the following inequalities are verified for an above-rated operating point:

$$h_t\frac{\partial \tau_a}{\partial v} - k_{\tau_g}N_g \geq 0 \tag{52a}$$

$$\frac{\frac{\partial \tau_a}{\partial \beta}}{\frac{\partial F_a}{\partial \beta}} > 0 \tag{52b}$$

$$\frac{1}{2\sqrt{K_t J_t}}\left(D_t + h_t^2\frac{\partial F_a}{\partial v}\right) \leq \zeta_{plt} \tag{52c}$$

First inequality comes from eq. (30) (notice that in Section 3, it is considered $\tau_g = 0$). Third inequality is a consequence of the assumption that $\zeta_{plt}$-fixed strategy aims at increasing the damping of the platform. This implies that:

$$\frac{\partial}{\partial_{\zeta_{plt}}} |A_{2,4}| > 0 \tag{53}$$

meaning that the first order coupling between platform dynamics and rotor dynamics will increase when $\zeta_{plt}$ increases if $\zeta_{plt}$-fixed strategy is applied. Thus, if the characteristic time of platform dynamics is small enough, the equation truncated at first order is valid and it is to be expected that, at least for some tunings of the PI controler, $\zeta_{plt}$-fixed strategy would increase rotor speed variations.

## 3 Numerical tests with time domain simulations

In this section, it is analysed how the new control strategy described in the previous section affects platform and rotor dynamics, and especially the impact on tower loads and rotational speed.

The simulation tool used is OpenFast v2.4.0 (https://github.com/OpenFAST/openfast) and the FOWT model considered is the IEA 15 MW wind turbine mounted over the UMaine VolturnUS-S semi-submersible floater (Allen et al., 2020). Initially simple constant wind and monochromatic waves are tested in order to verify the analytical developments of the previous section. Then a test case more representative of the industrial design of FOWT is considered by testing a DLC1.2. For the latter, simulation consider only 1 seed of 3600 seconds with aligned wind and irregular waves. For this time simulation, this is statistically equivalent to 600 seconds and 6 seeds.

### 3.1 Setup of the controller

The controller strategies are implemented in the ROSCO environment (ROSCO, 2021), modifying the existing pitch control. The rest of the controller remains basically the same. Values of $k_P$, $k_I$, $k_\beta$ are continuously updated following the explicit expression given in equations (37) and (47), then low-pass filtered. After some tests, for all the considered simulations, $\zeta_{rot} = 0.6$ and $\nu_{rot} = 0.01$ are chosen for the PI controller's tuning. This choice ensures that most of the wave spectrum (which peaks at $T \approx 11\ s$, i.e. $\nu \approx 0.57\ rads^{-1}$) and platform dynamics natural angular frequency ($\nu_{plt} \approx 0.22\ rad.s^{-1}$) fall outside of $G_{rot}$ pass-band. This strategy is known in literature as a *detuning strategy*. We consider here a very simplified version of the *detuning strategy* since the tuning of the PI controller is not the main object of this article. Since the platform pitch control ($\beta_{comp}$) has a damping effect mainly on the frequencies close to the natural frequency of the platform, its tuning can reasonably be considered as independent from the tuning of the PI controller, which is already well assessed in literature. The blade pitch saturation defined in ROSCO is switched off in order to better observe the effect of the platform pitch control strategies. Moreover, $\tau_g$-compensation is not assessed in this section, as it has already been studied by Stockhouse et al. (2021). Hence, $k_{\tau_g} = 0$ hereafter. The interactions between blade pitch saturation, or $\tau_g$-compensation, and the proposed blade pitch controller strategy should be investigated in future works.

## 3.2 Still wind and monochromatic wave

For the still wind and monochromatic wave condition, $\zeta_{plt}$-fixed strategy is compared to $k_\beta = 0$, i.e. the detuning strategy. Two $\zeta_{plt}$ are tested: $\zeta_{plt} = 0.1$ and $\zeta_{plt} = 0.25$ [3]. Thus, the platform is expected to behave like an under-damped second order oscillatory system. Table 4 states external conditions for test cases with still wind and monochromatic wave. The platform is subjected to a monochromatic wave of period $11s$ (a representative value for the fundamental period of a wave spectrum) and $28.75s$ (the natural period of the platform). The wave height is the same but in the corresponding linear model the resulting input's ($\tau_{wave}$) amplitude is different (as it also depends on the wave period). Hereafter, results are plotted over time are drawn for a 100 seconds time interval in a simulation on a long period of time, so that the operating point is reached. When necessary for a better understanding, results are reported for a longer interval.

**Table 4.** Environmental conditions for the numerical test cases.

| case | wind speed $V$ $(ms^{-1})$ | wave period $Tp$ $(s)$ | wave height $H_w$ $(m)$ |
|------|------|------|------|
| (1) | 11 | 11 | 1.5 |
| (2) | 11 | 28.75 | 1.5 |
| (3) | 22 | 11 | 1.5 |
| (4) | 22 | 28.75 | 1.5 |

This test considers fixed operating points, thus, $k_P$, $k_I$, $k_\beta$ have almost fixed values. Table 5 gives the mean value of $k_\beta$ for test cases (1) to (4). Cases (1), (2) and cases (3), (4) are gathered together as they use the same mean value of $k_\beta$.

**Table 5.** compensation gain ($k_\beta$) corresponding to Table 4.

| case | $\zeta_{plt} = 0.10$ | $\zeta_{plt} = 0.25$ | reference |
|------|------|------|------|
| (1) and (2) | $k_\beta = -8.6$ | $k_\beta = -42.7$ | $k_\beta = 0.0$ |
| (3) and (4) | $k_\beta = -7.4$ | $k_\beta = -34.8$ | $k_\beta = 0.0$ |

[3]For the readers more used to quality factors, the corresponding values are: $Q = 5$ and $Q = 2$, respectively

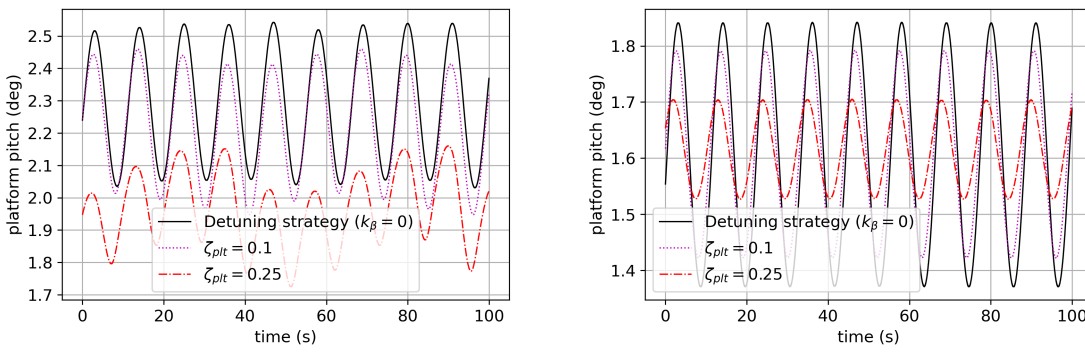

**Figure 9.** Platform pitch $\Phi$ (deg) for a monochromatic wave of period 11s (test case (1) on the left and (3) on the right).

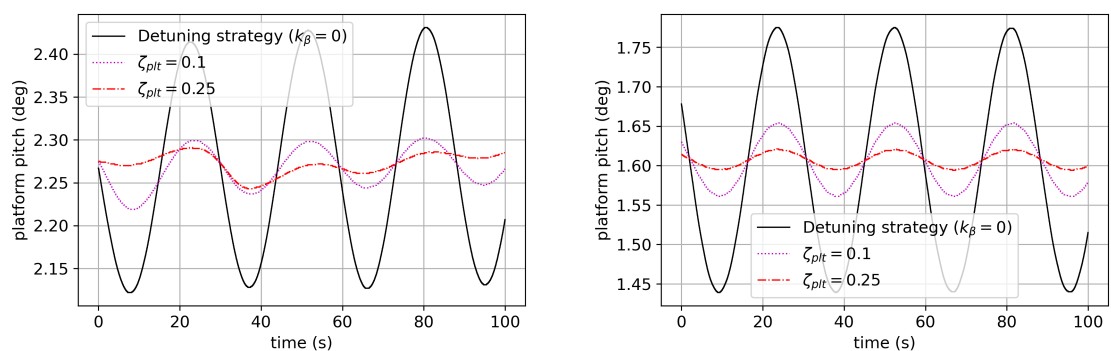

**Figure 10.** Platform pitch $\Phi$ (deg) for a monochromatic wave of period 28.75s (test case (2) on the left and (4) on the right).

Figures 9 and 10 show the forced oscillations of the platform when it is subjected to several test cases (see Table 4) Increasing $\zeta_{plt}$ reduces platform oscillations, especially when the wave period is close to the natural period of the platform. Even though the system is much more complex in those simulations, this general behavior was forecast (see Section 2.5.1 and Figure 8) by the damping analysis on the two-dimensional linear model described in Section 2.4.2 . The damping effect is also shown in Figures 12 and 13 where the tower base reaction moment is plotted over time. For the monochromatic wave with period $11s$ the damping of the $\zeta_{plt}$-fixed strategy is not evident, while it becomes easily observable as soon as the input's frequency gets close to the platform's natural frequency.

Figures 9 shows a reduction in the amplitude of the platform oscillations and also in the mean value. This was unexpected. It is motivated by the difference in the mean value of the blade pitch. In Figure 11, one can see the high values in the peaks for $\zeta_p lt = 0.25$ with respect to the other curves. The minima are comparable, then the mean is higher. This leads to a lower thrust force in average which reduces the mean value of the platform pitch. The peaks in the blade pitch can be motivated by the combination of high demanded damping ($\zeta_p lt = 0.25$), with the proximity to the rated wind speed (the controller is on the

boundary of regions 2.5 and 3) and also the wave period. In fact, for case (2), where the wind speed is the same but the wave period is much higher, the mean value of the platform pitch does not change.

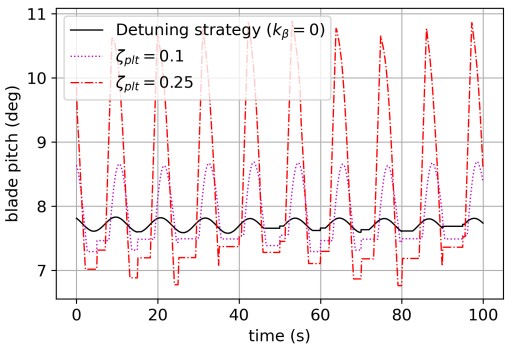

**Figure 11.** Blade pitch (deg) evolution over time for a monochromatic wave of period 28.75s (test case (1)).

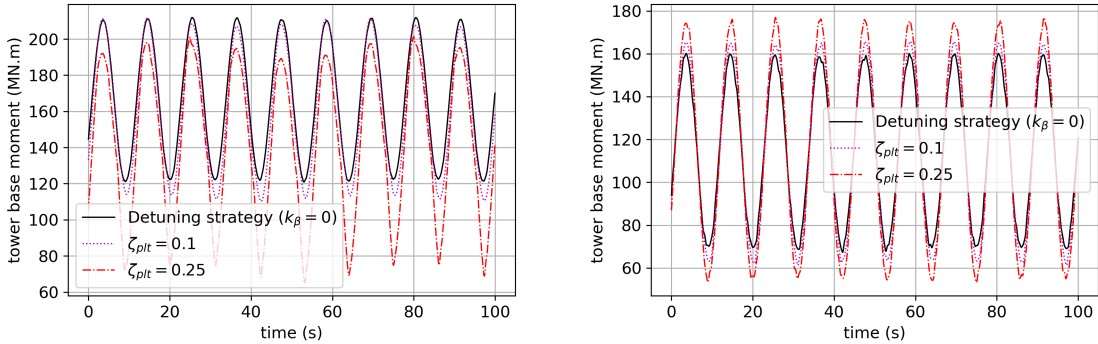

**Figure 12.** Tower base moment (MN.m) evolution over time for test case (1) on the left and (3) on the right.

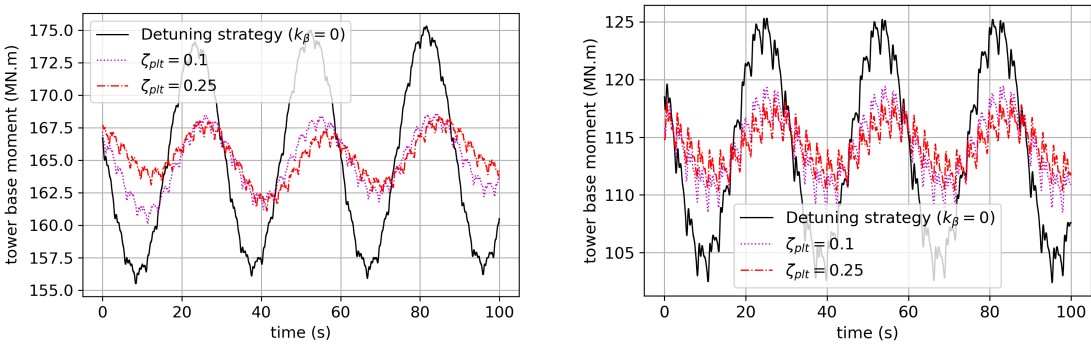

**Figure 13.** Tower base moment (MN.m) evolution over time for test case (2) on the left and (4) on the right.

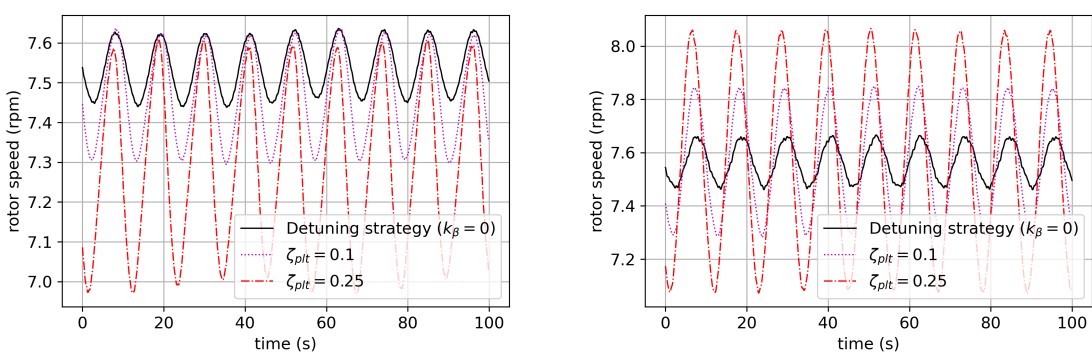

**Figure 14.** Rotor speed (rpm) evolution over time for test cases (1) on the left and (3) on the right

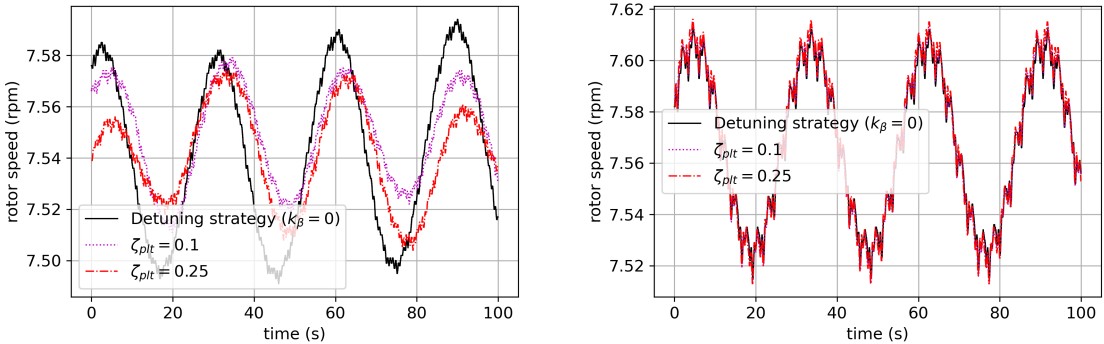

**Figure 15.** Rotor speed (rpm) evolution over time for test cases (2) on the left and (4) on the right.

$\zeta_{plt}$-fixed strategy's effect on rotor dynamics is not easily described by a second order linear equation: it involves the coupling between platform and rotor dynamics, which is analysed at first order in Section 2.5. From this analysis, $\zeta_{plt}$-fixed strategy was expected to increase the coupling between platform and rotor dynamics for a short characteristic time. In Figures 14 and 15 it can be observed that for a short characteristic time ($11s$) $\zeta_{plt}$-fixed strategy increases rotor speed variations, but not for a longer characteristic time, such as $28.75s$ (for which it behaves slightly better than $k_\beta = 0$ strategy).

To conclude this part of the tests: $\zeta_{plt}$-fixed strategy performs very differently depending on the oscillatory frequency of the platform:

- For angular frequencies $\nu \in I_{damped} = \left[ \frac{\nu_{plt,natural}}{\sqrt{2}} \ , \ \sqrt{2}\nu_{plt,natural} \right]$, $\zeta_{plt}$-fixed strategy is very effective when it comes to the damping of platform oscillations, as seen in Section 2.5 (cf. figure 8). The tests highlight that $\zeta_{plt}$-fixed strategy is reducing both tower loads and rotor speed variations in turbulent wind conditions.

- For angular frequencies outside the previous set, $\zeta_{plt}$-fixed strategy is less effective for damping platform oscillations. Tower loads reduction by $\zeta_{plt}$-fixed strategy is therefore barely visible, whereas rotor speed variations are actually amplified, especially when comparing this strategy to reference strategy.

## 3.3   DLC1.2 tests

The tests presented hereafter are more representative of what is typically done during the design or verification of offshore
wind structures. They are inspired by DLC 1.2, for normal power production in normal turbulence and normal sea state, as described in the IEC standards. This kind of load case aims at assessing the fatigue design criteria.

    Kaimal's turbulence model is considered following IEC 61400 v.3 for a wind turbine of turbulence type B, for average wind speeds ranging from $4 \, ms^{-1}$ to $24 \, ms^{-1}$, as described in Table 6. The wind box is generated by the TurbSim tool developed by the NREL. For the waves, JONSWAP distributions are considered with a significant wave height of $H_s = 1.5 \, m$, wave period
$T_p = 11.0 \, s$ and $\gamma = 2.0$. Wind and waves are considered to be aligned in the same direction. All the degrees of freedom of the floating platform are allowed, including the surge motion. In other words, the numerical twin reproduces the actual motion of the FOWT, according to the accuracy of the chosen model. Tower and blades are fully deformable.

**Table 6.** Environmental conditions for DLC 1.2.

| Time sim [$s$] | w.speed [$m.s^{-1}$] | w. condition | $Tp$ [$s$] | $H_s$ [$m$] | $\gamma$ | waves dir. |
|---|---|---|---|---|---|---|
| 3600 | $4.0 - 24.0$ | Normal turbulence B | 11.0 | 1.5 | 2.0 | co-linear |

    As with the previous test cases, the $\zeta_{plt}$-fixed strategy is compared to the detuning strategy, i.e. $k_\beta = 0$. Another term of comparison is considered in this section. The $\zeta_{plt}$-fixed strategy considers a $k_\beta$ that adapts to the wind and evolves during the
470 simulation. This is different from what is implemented in ROSCO controller (ROSCO, 2021) which considers one $k_\beta$, tuned only once for a given FOWT and applied for every wind speed. This strategy is considered in this section as a second term

of comparison. For the latter, the $k_\beta$ value is set to $-9.35$, i.e. the value tuned for this FOWT in https://github.com/NREL/ROSCO/blob/main/Test_Cases/IEA-15-240-RWT-UMaineSemi/DISCON-UMaineSemi.IN.

The level of damping imposed to the platform is $\zeta_{plt}$-fixed $= 0.1$. This value is found to be the most interesting to be tested for this floater and WTG configuration. Other tests with higher values of imposed damping show less interesting results. The choice of the right $\zeta_{plt}$ for each FOWT system is important and may require several iterations before a conclusion can be reached.

Table 7 resumes the strategies considered for the benchmark in this section, resuming the difference in the choice of the platform pitch compensation.

**Table 7.** strategies considered for the DLC1.2 tests.

| Strategy name | $\nu_{rot}$ | $\zeta_{rot}$ | $k_\beta$ |
|---|---|---|---|
| detuning | 0.01 | 0.6 | 0.0 |
| $\zeta_{plt}$-fixed | 0.01 | 0.6 | eq. (47) with $\zeta_{plt} = 0.1$, adapting to wind |
| $k_\beta$-constant | 0.01 | 0.6 | $-9.35$, not adapting to wind |

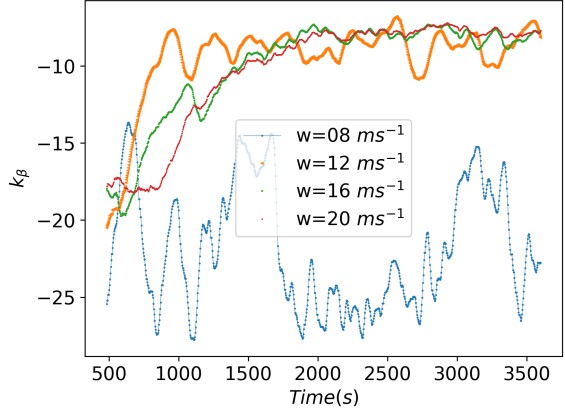

**Figure 16.** Evolution of $k_\beta$ for some of the simulations. On left, $k_\beta$ is obtained by eq. (47) and

Figure 16 shows how $k_\beta$ evolves for some cases of the simulation pool. The platform pitch compensation takes place also for the below-rated wind speeds, where it is the only source of pitch control. The behaviour in below-rated wind speeds is more dynamic than in above-rated wind speeds, where the floating feedback is more stable. As remarked in Section 2.5, the $k_\beta$ for the $\zeta_{plt}$-fixed is a negative value.

From the time series of the tower bottom moment, a rainflow algorithm is used to count the cycles following ASTM normative (ASTM, 2017). The Design Equivalent Load (DEL) is obtained by using a Wohler's curve with a single slope of exponent

$m = 3.0$. Platform pitch, power, rotor speed, blade pitch, tower load and tower DEL results of the simulations for the comparison are resumed in Figure 17. The $\zeta_{plt}$-fixed strategy reduces platform pitch motion for all wind speeds when compared to the detuning strategy, and it gives comparable or better results with respect to the $k_\beta$-constant strategy. . The three strategies give comparable results when looking at power generation, rotational speed and mean values of blade pitch. However, for the 20 $ms^{-1}$ wind speed, $k_\beta$-constant shows an over-speed in the rotor (max rotor speed) and a higher value in the platform pitch standard deviation. This coupling among platform pitch motion, rotor speed and blade pitch affects the DEL at the tower base. It helps show the benefit of considering a $k_\beta$ that adapts to the wind speed and the different wind turbine operations while considering a single $k_\beta$ for the FOWT system has the potential to result in irregular performances.

Around rated wind speed, the $\zeta_{plt}$-fixed strategy reduces loads in the tower and fatigue DEL when compared to the other two strategies. For high wind speeds, the gain is less evident, especially when comparing to the $k_\beta$-constant strategy, which has a lower DEL for 24 $ms^{-1}$. In average there is a gain around 15 % of the DEL with respect to the detuning strategy. This gain is less evident when compared to the $k_\beta$-constant strategy. Nevertheless $\zeta_{plt}$-fixed strategy seems to perform better around rated wind speeds and it gives a more homogeneous performance than $k_\beta$-constant strategy. Table 8 shows, for the 10 $ms^{-1}$ case, a deeper comparison by reporting the statistics of the quantities of interest extracted from this simulation. For this wind speed, an extract of time-series for some outputs of interest is shown in Figure 18. It is intended to give a better view of the damping effect given by the $\zeta_{plt}$-fixed strategy.

Figure 19 reports a deeper analysis of the fatigue damage. In fact, the stress in the tower bottom section is obtained by considering the design proposed by UMaine in (Allen et al., 2020). Then, an offshore Wohler's curve is considered with two slopes in the log-log domain: $m = 3.0$ for loads with less than 1.0 million cycles and $m = 5.0$ for loads with higher number of cycles. Those are typical values proposed by DNV for offshore steel structures. This analysis leads to obtain an estimation of the 25 years damage at the tower bottom. The gain is much more evident than the DEL. This is due to the second slope, $m = 5.0$, which amplifies the changes in the load amplitudes. Offshore WTG in production are mostly subjected to a very high number of cycles of small amplitudes. This figure shows also the effect of the turbulence on the fatigue. In fact, looking at the detuning strategy, up to 12 $ms^{-1}$, the shape of the damage distribution follows the one of the thrust curve. However, since the turbulence is a percentage of the average wind speed, from 16 $ms^{-1}$, the damage starts increasing again. In general, the $\zeta_{plt}$-fixed strategy is more adapted than the other two strategies for the fatigue of the structures. It demonstrates a reduction of approximately 20 % in the cumulative damage compared to $k_\beta$-constant strategy, and of approximately 30% compared to the detuning strategy.

A specific fatigue analysis of the pitch bearing is realized by following (Shan et al., 2021), where three methods to evaluate the fatigue of the pitch bearing are compared leading to comparable results. The second method is implemented here in order to quantify the increment in the pitch bearing fatigue caused by the $\zeta_{plt}$-fixed and $k_\beta$-constant strategies with respect to the detuning strategy.

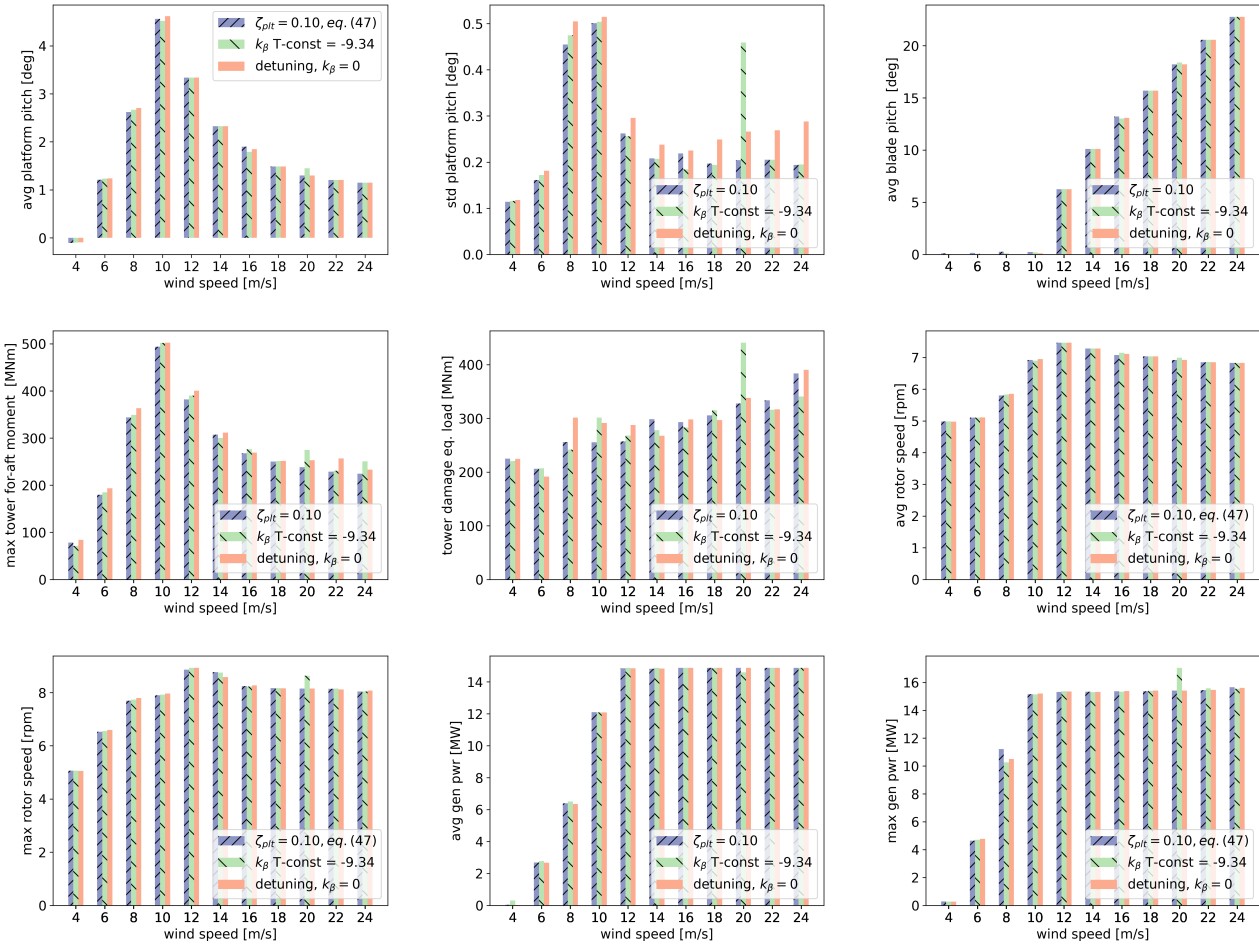

**Figure 17.** Comparison results for the DLC1.2 for the UMaine floater with IEA15MW WTG. Outputs show statistics for platform pitch (average and std); blade pitch; tower bending moment, max and damage equivalent load; rotor speed (average and max) and generator power (average and max).

The bearing life is inversely proportional to the cube of the bearing loading. From the overturning moment acting on the bearing, the equivalent loading at $N$ revolutions of the pitch bearing is given by:

$$M_{eq} = \left( \sum_i \frac{\Delta\beta_i \, M_i^3}{N} \right)^{1/3} \tag{54}$$

where $i$ is the time step of the simulation. The discrete integral considers the product of the time series of the overturning moment $M_i$ and the blade pitch variation $\Delta\beta_i$, over the entire simulation. To take into account the fact that, for each wind speed, the mean blade pitch is different, the 90 degrees of the pitch range are divided in 30 sectors, each one corresponding to a different zone of the bearing. This corresponds to consider a tooth function in the integral of eq. (54) and it is well explained

**Table 8.** Statistics for results concerning the case with mean wind speed at $10 \ ms^{-1}$. For each quantity of interest, there is the comparison of the minimum, maximum, mean and standard deviation values produced by the $\zeta_{plt}$-fixed, $k_\beta$-constant and detuning control strategies.

| | $\zeta_{plt}=0.1$ | $k_\beta=0$ | $k_\beta=-9.35$ | $\zeta_{plt}=0.1$ | $k_\beta=0$ | $k_\beta=-9.35$ | $\zeta_{plt}=0.1$ | $k_\beta=0$ | $k_\beta=-9.35$ | $\zeta_{plt}=0.1$ | $k_\beta=0$ | $k_\beta=-9.35$ |
| | min | min | min | mean | mean | mean | max | max | max | st.d. | st.d | st.d. |
|---|---|---|---|---|---|---|---|---|---|---|---|---|
| PtfmPitch [deg] | 2.84 | 1.09 | 2.87 | 4.55 | 4.56 | 4.56 | 5.62 | 5.91 | 5.69 | 0.51 | 0.52 | 0.51 |
| TwrBsMyt [MNm] | 165 | 81 | 167 | 366 | 371 | 368 | 494 | 505 | 502 | 43.2 | 46.8 | 42.0 |
| GenPwr [MW] | 7.85 | 5.49 | 8.05 | 12.1 | 12.0 | 12.1 | 15.2 | 15.2 | 15.1 | 1.29 | 1.47 | 1.31 |
| BldPitch [deg] | 0.0 | 0.0 | 0.0 | 0.218 | 0.176 | 0.178 | 6.65 | 8.33 | 6.90 | 0.394 | 0.729 | 0.424 |
| RotSpeed [rpm] | 5.23 | 5.34 | 5.24 | 6.90 | 6.92 | 6.90 | 7.90 | 8.00 | 7.93 | 0.61 | 0.60 | 0.59 |

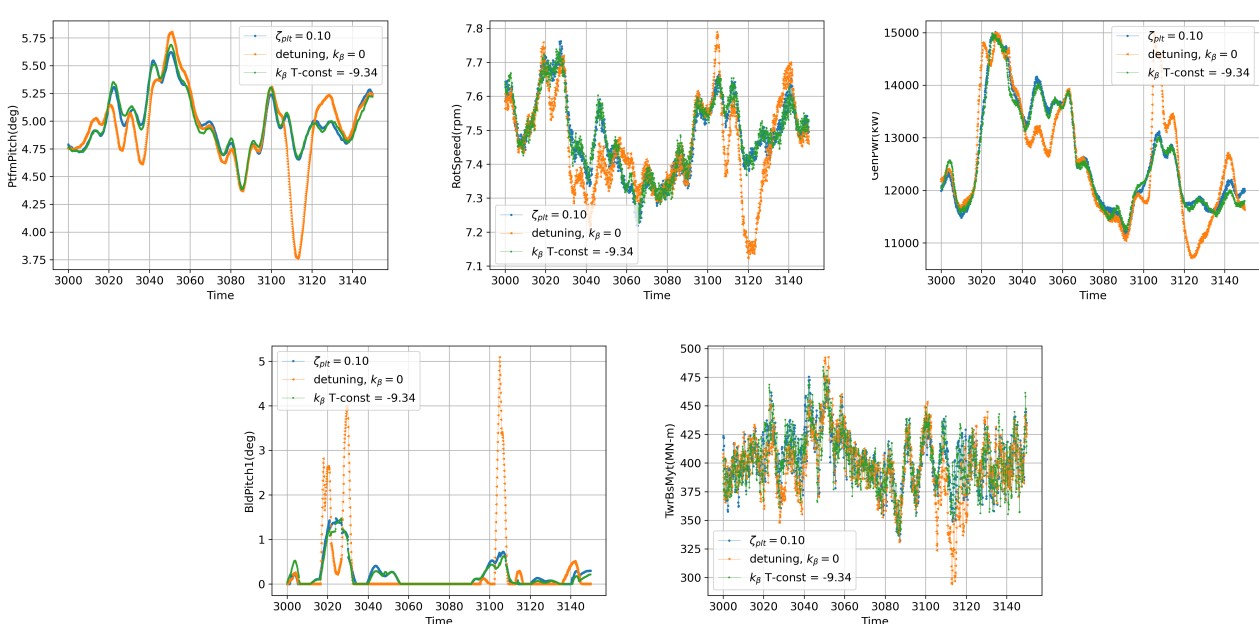

**Figure 18.** Extract of the time series concerning the case with mean wind speed at $10 \ ms^{-1}$. The $\zeta_{plt}$-fixed, $k_\beta$-fixed and the detuning control strategies are compared in platform pitch; rotor speed; generator power; blade pitch; and tower bending moment .

in Figure 11 of (Shan et al., 2021). Figure 20 reports the damage equivalent loading $M_{eq}$ given by the three strategies . Overall, the fatigue of the bearing is increased by introducing a platform pitch compensation when compared to the detuning strategy. This is an aspect of the control strategy to be considered and it is an axis of improvement for future works. For $k_\beta$-constant, the instability problem taking place at $20 \ ms^{-1}$ between blade pitch and platform dynamics is underlined by a strong increase in the fatigue of the pitch bearing.

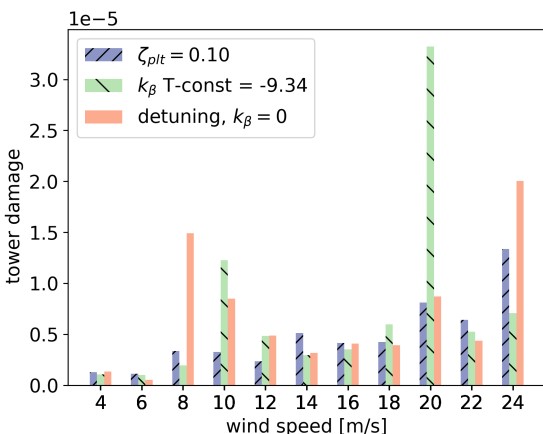

**Figure 19.** Fatigue cumulative damage at tower bottom by using rainflow counting and linear Miner's rule. The damage is obtained considering the tower base design proposed by the UMaine, a Wohler's curve bi-linear with $m = 3.0$ up to $10^6$ cycles and $m = 5.0$ after, as proposed by DNV for Offshore steel. The probability of occurrence of each wind is equal, without any weibull distribution. $\zeta_{plt} = 0.1$ reduces for about 20% the overall cumulative damage when compared to $k_\beta$-constant strategy and about 30% with respect to detuning strategy.

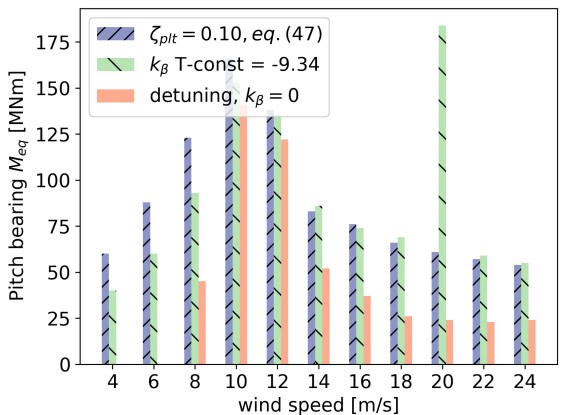

**Figure 20.** Damage equivalent loading at $N$ revolutions of the pitch bearing given by the three strategies (defined by eq. (54)).

## 4 Conclusions

The first part of this paper presents the analysis of the NMPZ related to the system of equations describing the dynamics of a floating offshore wind turbine (FOWT). The equation of the rotor dynamics and the one of the platform dynamics are analysed in the complex domain to explicitly derive the conditions leading to their respective NMPZs. One of those NMPZs, i.e. the instability given by the blade pitch on the rotor dynamics, is already known in literature and a compensation already exists to

avoid it. The other one, i.e. the instability given by the blade pitch to the platform dynamics, is a novelty in the community. The effects of the NMPZs are analysed on two analytical examples: at the beginning both $\omega$ and $\dot{\phi}$ always converge to the right solutions just after the first steps. When both NMPZ conditions are not verified, those tendencies do not change. However, when the $\dot{\phi}$-NMPZ is verified, $|\omega|$ becomes so big that $\dot{\phi}$ jumps into unexpected values without converging to the expected solution. Similarly, when the $\omega$-NMPZ condition is verified, $|\dot{\phi}|$ becomes so big that $\omega$ oscillates before converging to the expected

solution. NMPZs can cause important shifts and unexpected behaviors for both $\omega$ and $\phi$. For those examples, the position of poles and zeros of the transfer function in the complex domain is analyzed. The beneficial effect of the compensation for the $\omega$-NMPZ is shown by plotting the displacements of the poles and zeros from the right to the left part of the complex domain.

In the second part of the paper, the damping analysis is further investigated while proposing a new strategy control for FOWT, named $\zeta_{plt}$-fixed. This strategy is based on a compensation parameter $k_\beta$, which is proportional to the platform pitch

velocity. It considers the coupling between the rotor dynamics and the floating platform dynamics. The idea behind this control strategy is to activate the blade pitch to damp the platform motions. An explicit expression linking $k_\beta$ to $\zeta_{plt}$ (damping ratio imposed to the platform) is obtained by deriving a second order filter from the equation of the platform dynamics.

This is different with respect to existing platform pitch compensation strategies which aim to decouple rotor and platform dynamics. The difference is underlined by the values of $k_\beta$, which is negative for the new control strategy, while it is positive for

the ones existing in literature. For each FOWT system, some iterations are necessary in order to find the optimum value for $\zeta_{plt}$. The performances of the $\zeta_{plt}$-fixed strategy are tested analytically and numerically by considering an OpenFAST numerical twin of the Umaine IEA15MW FOWT. For a test representative of the DLC1.2, the $\zeta_{plt}$-fixed strategy allows to reduce the loads at the tower foundation interface for all the considered wind speeds, without significant losses in terms of power production. When compared to a strategy considering a constant platform pitch compensation, it reduces fatigue damage by about 20 %,

underlining the benefit of considering a $k_\beta$ that adapts to the wind speed and the different wind turbine operations. The damage analysis shows a remarkable gain in terms of fatigue lifetime of the structure but also an increase in the use of the blade pitch bearing.

This work highlights the importance of defining proper controller strategies for FOWT in order to reduce loads on the structure or improve the performance. Accordingly, it is useful in helping the industry to achieve the objective in terms of

LCOE reduction.

*Author contributions.*  Matteo Capaldo has contributed for the original idea of the new control strategy, the development of the numerical twin and the numerical tests and he is the main contributor for the paper editing. Paul Mella developed the mathematical framework, he has contributed for the original idea of the new control strategy and he has contributed for the paper editing

*Competing interests.*  The authors declare there are not competing interests in this work.

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
