# Peer review of "Damping analysis of Floating Offshore Wind Turbine (FOWT): a new control strategy reducing the platform vibrations"

_Wind Energy Science, 2022_

## Referee Comment (RC1)

510

[referee-annotated manuscript omitted]

---

## Author Comment (AC1)

**Answers to reports on wes-2022-109**

Paul Mella and Matteo Capaldo

February 10, 2023

**Answer to anonymous referee 1**

1. **Comparison about ROSCO platform pitch controller strategy and the proposed platform pitch controller strategy**: The main difference in the two approaches can be remarked at lines $305-310$. As it can be remarked, the two ways to define the platform pitch compensation $k_\beta$. ROSCO derives the parameter imposing the rotor dynamics and the platform pitch dynamics to be decoupled at the first order. In other words, the effect of relative wind generated by the platform pitch dynamics is, at first order, compensated by feathering. The strategy proposed in this new paper aims at taking advantage of the blade pitch control influence on the platform pitch dynamics in order to introduce an extra term in the second order dynamics equation of the platform pitch. Thus the second order dynamics equation has an explicit form involving a damping ratio $\zeta_{plt}$ whose value one can explicitly define.

   On can, then, notice that the two formulas to define $k_\beta$ for ROSCO strategy and the proposed strategy are different. Also numerically, they lead to values that are opposite in sign.

   Indeed, $\frac{\partial F_a}{\partial \beta} < 0$ (for an above-rated operating point) and therefore using inequality (51), we find that

   $$k_\beta = \frac{1}{h_t \frac{\partial F_a}{\partial \beta}}\left(2\sqrt{K_t J_t}\zeta_{plt} - D_t - h_t^2 \frac{\partial F_a}{\partial v}\right) < 0$$

   On the other hand, ROSCO strategy, as it is defined in (Abbas, 2022) or (Sotckhouse, 2021) derives from the equation $A_{2,4} = 0$ where $A$ is the matrix defined in (17) ((Abbas, 2022) introduces that same matrix) and expresses the platform pitch control coefficient as

   $$k_\beta = -h_t \frac{\partial \tau_a}{\partial v} / \frac{\partial \tau_a}{\partial \beta} > 0$$

   It is negative since $\frac{\partial \tau_a}{\partial v} > 0$ and $\frac{\partial \tau_a}{\partial \beta} < 0$ (for an above-rated operating point). Notice that in (Abbas, 2022), $\beta_{comp}$ is defined as in (Stockhouse, 2021) but with the convention $\beta_{comp} = k_{float}\dot{\phi}$ so that $k_{float} = -k_\beta = h_t \frac{\partial \tau_a}{\partial v} / \frac{\partial \tau_a}{\partial \beta}$ is negative, but this is just a question of conventions. If one takes the same convention, the sign is actually switched in our formula.

   About the comparison on the numerical tests, we tried to compare to ROSCO during the work in two ways: by using the ROSCO controller as downloaded from github and also by implementing the equations explained in Abbas2022. In both cases, the obtained results not in line with the ones found in Abbas2022. Hence, we concluded that we missed some other parameters for those simulations. It is much better to compare to a strategy with $k_{beta} = 0$. This term of comparison has not been well presented in the paper, but it corresponds to a "detuning strategy". In fact, to find the right $\nu_{rot}$ and $\zeta_{rot}$ of equations (36) at line $274 - 279$, a lot of simulations have been tested in order to find the best values for this platform and this wind turbine generator. However, if necessary one can compare the strategies by looking at the article Abbas2022 which considers the same numerical test as the one considered in this article.

2. line 50: comment accepted and introduced in the next paper version.

3. line 64: Lenfest2020 linearizes the $k_\beta$ (called $k_px$ in his article) and the authors calibrate a scheduling of the values of $k_\beta$ by testing many numerical values, instead of making an analytical studies on the platform damping to define the parameter. The proposed strategy has an explicit form involving a damping ratio $\zeta_{plt}$ whose value one can explicitly define to obtain the right parameter.

4. Line 75, the hinge point of the platform is the COG of the platform. An image will be added to make it clear.

5. 91, comment accepted and introduced in the next paper version ($\tau_w$ replaced by $\tau_{wave}$).

6. Line 96: comment accepted and introduced in the next paper version.

7. Line 102, comment accepted: the sentence is removed in the next paper. It does not add any information.

8. Line 121, comment accepted and introduced in the next paper version.

9. Line 125, "relative speed" replaced by "infinitesimal speed" $\omega = \Omega - \Omega_r$.

10. Line 125, Yes, it is indeed the onshore standard blade pitch control.

11. Figure 1: we'll modify the scheme to remove $\dot{\Phi}_r = 0$.

12. Line 153: "closed loop" will be added to the sentence.

13. Line 155: this is the *Laplace* domain. It will be clarified in the next version.

14. Line 161: the size of the equations with matrix is too big to write it in the article. It could be done in an Annex, if necessary.

15. Line 166: corrected

16. Line 174: corrected

17. Line 174: corrected

18. Line 167: corrected

19. Line 185: "the amplitude" will be replaced by "the importance" to make it clear

20. Line 191: "with respect to the platform pitch at the operating point" will be added

21. Line 193: we propose to replace the sentence by "When the equation is verified, it means that: $\tau_a$ is more sensitive to blade pitch than rotational speed and $F_a$ is more sensitive to omega than blade pitch. Therefore, by increasing blade pitch $\omega$ increases and, then, occasions $F_a$ to decrease. Then, $\phi$ increases."

22. Line 200 Table 1. the values are determines arbitrarily as examples, to ensure the appearance of NMPZs. This is done in the purpose of a pedagogic example to show the effects of the NMPZs. The values are purely numerical. There are not so far from the coefficients related to the IEA15MW wind turbine because we started from those physical values and we found the right coefficients, in the order of 10% or 20%, to make the system unstable. The numerical section focuses on more realistic FOWT tests.

23. Line 200. Caption clarified by: "Platform pitch ($\phi$) and rotor speed ($\omega$) responses to a blade pitch ($\beta$)-step input ..."

24. Line 220. A graphic visualization of the roots of G(s) will be added. We'll show how those roots move when $m_{\tau_g}$ varies, and that will explain its utility. (Stockhouse, 2021) introduces $m_{\tau_g}$ and explains that the choice of $m_{\tau_g} = 1$ avoids any eventual NMPZ, but is not always possible because of $\tau_g$ saturation. The stability margin will be discussed.

25. 232: for the NMPZ $\omega --> \phi$ a solution is proposed by (Stockhouse, 2020) with the introduction of the $m_{\tau_g}$. For the NMPZ $\beta --> \phi$, at our knowledge it is the first time to be analysed. This paper does not investigate a solution.

26. Line 259: the coupled system is 4x4 fully populated. It is very hard to be diagonalized. We tried to solve it by using symbolic software languages. However, it didn't come to an outcome, it didn't give anything usable in practice. It has been chosen to continue on 2x2 simplified systems. In this case, damping expressions are explicit. Modal dampings could be expressed but they they should be very similar to the one already expressed because the system has only 2 dofs.

27. Line 273: Indeed, this is the same equation as for a rotor with a fixed nacelle.

28. Line 283: Actually here we are not in open-loop condition. The hypothesis $Kp = Ki = 0$ is to arrive to a 2x2 system and the control is ensured by $k_\beta \neq 0$

29. Line 322: in text we'll add "disturbance and open-loop" input.

30. Line 352: $k_{\tau_g} = 0$ has already been studied by "Stockhouse" and it was not interesting to repeat the same study

31. Line 360: Quality factor is defined by
$$Q = \frac{1}{2\zeta}$$

As some readers might be more used to work with quality factors instead of damping ratios, we thought it was a good thing to give the quality factors corresponding to the damping ratios. It is not mandatory for the comprehension of the paper.

32. Line 366: We'll add in the text: "corresponding to Table 4" as suggested.

33. Line 367: When the platform pitch rate is compensated, the Thrust changes because of the change in the blade pitch (induced by $k_\beta$) and the mean platform rotational position change. In fact the system slightly shifts to another operating point.

34. Line 367 (Figure 6): When comparing results from section 3. with Figure 5, one should keep in mind that in in section 3, the system is way more complex: Figure 5 corresponds to a 2-dimensional state-space model. Moreover, what is underlined by Figures 6 and 7 is the comparison between the damping of an input of period 11s and an input of period 28.75s. In figure 6, the gain is much lower than in figure 7 and we can, then, see a similar trend to the one of Figure 5. When comparing Figures 6 and 7, one should also keep in mind that the chosen wave conditions have same height, but this does not mean that the input's amplitude is the same. We'll try to make this clearer in the paper.

35. Line 370: Figures 8 and 9 will be better explained.

36. Line 374: section 2.5 will be replaced by equation (50)

37. Line 399: Yes, $k_\beta$ changes with time. A filter is also applied to change it smoothly since it focuses on low frequencies and it is useless to have high frequency changes.

38. Line 401: "section" added in the text.

39. Line 406: The mean value of the blade pitch changes because the increase in blade pitch variations makes the rotational speed increase. Hence in order to have this variations of the rotational speed, the mean value has to slightly decrease in order to not overpass the rated rotational speed.

40. Line 415. We correct the text by removing sentence "It is interesting .."

41. Line 424 (Table 7). This question is related to the one at Line 406. Please, refer to the previous answer.

42. Line 439. We'll explain variables in the caption.

43. Line 446. We'll replace in the text "those tendencies ..." by "they keep converging to the right solution".

---

## Author Comment (AC2)

**Answers to reports on wes-2022-109**

Paul Mella and Matteo Capaldo

February 10, 2023

**Answer to anonymous referee 2**

1. Lines 25 and 29. In the text we'll replace "oscillating stability" by "oscillating steady-state"

2. Line 27. We'll correct in the text

3. Line 37. We'll correct in the text

4. Line 40. We'll replace "However the platform pitch damping analysis is not investigated and the link with the compensation parameter is not given " by "(Lenfest et al., 2020) investigates the (coupled) platform pitch damping and parameter tuning with a purely numerical approach. In the present work, we propose an explicit formulation for the tuning parameter related to this damping which depends on the system properties."

5. Line 54 (and other occurrences). We'll add Fischer2013 to our bibliography. We'll cite it with Stockhouse for the previous works dealing with the platform pitch compensation

6. Line 107 and others. We'll correct in the text.

7. Line 120. We'll delete " $= k_I(\theta - \Omega_r)$". It didn't add info.

8. Line 129. We'll correct in the text.

9. Line 133. We'll correct in the text.

10. Line 142. We'll correct in the text.

11. Lines 165 - 175. We'll add this reference for the NMPZs definition and comprehension: Hoagg, Bernstein, 2007, IEEE Control Systems Magazine.

12. Line 177. We'll correct in the text.

13. Line 180. A graphic visualization of the roots of G(s) will be added. In particular, we'll show how those roots move when $m_{\tau_g}$ varies, and that will explain its utility. The stability margin will be discussed.

14. Line 185. We'll add the definition of WTG in the text.

15. Tables 1, 2, 3. We'll correct the errors in the units.

16. Tables 1, 2, 3. The values are determines arbitrarily as examples, to ensure the appearance of NMPZs. This is done in the purpose of a pedagogic example to show the effects of the NMPZs. The values are purely numerical. There are not so far from the coefficients related to the IEA15MW wind turbine because we started from those physical values and we found the right coefficients, in the order of 10% or 20%, to make the system unstable. The numerical section focuses on more realistic FOWT tests.

17. Figure 2. At lines 184 - 199, we explain intuitively how equation (25) can be interpreted and how it implies a flipped sign of the steady-state response. For more details at that subject, (Hoagg, Bernstein, 2007) is very clear. We will add this reference in our bibliography.

18. Figures 2, 3, 4. We'll add units for the y-axis ($\phi$ is in $rad$ and $\omega$ is in $rad.s^{-1}$). The beta-step input is of size $0.02 rad$. This precision will be added in the caption.

19. Line 206. for equation 28., we'll cite Stockhouse and Fischer.

20. Line 207-225. We'll give more details about $m_{\tau_g}$, including its dependency on $D_t$. For this, the graphical representation of zeros and poles will be useful.

21. Line 215. Indeed, $d\beta_{comp} = -k_\beta \dot\phi$! (Of course this is just a typo error and doesn't affect the analytical results.)

22. Line 221. (Stockhouse, 2021) discusses equation (29) : the parameter $m_{\tau_g}$ introduces a possible saturation of generator torque ($\tau_g$) control. We will rephrase this in order to make it clear.

23. sec.2.3.3. This section will be improved with the poles and zeros analysis.

24. Line 233. Gain scheduling approach defines a set of "set points". For each member of if, it provides a different set of pre-computed gain parameters. The hypothesis is that the system is linear close to the set point. Our approach make the same hypothesis. However, we don't need to pre-compute a set of gain parameters for each operational set point because the gain parameters are computed by the explicit formulas (equation 37). This explanation will be reformulated and added in the text for a clear understanding.

25. Line 238. We'll replace "negative damping" by "NMPZ phenomena (combined with the closed loop control)".

26. Non, generator torque parallel compensation is not implemented in the simulation because not necessary.

27. Line 328. Equation (51) will be separated into equation 51.a, 51.b, 51.c.

28. It would be good to have a section analyzing which NMPZ is verified and for which operating point, but I am not sure how we can obtain precise values for the partial derivatives that appear in the NMPZ conditions.

29. Line 352 and 354. For $k_P$ and $k_I$, many preliminary simulations, without the floating compensation, have been performed in order to obtain a set of values which works already very well for this system, without using any $k_\beta$. Those values of $\zeta_{rot}$ and $\nu_{rot}$ gave the best results, which might also be due to our choice to switch off blade pitch saturation (choice motivated to better observe the effect of our platform pitch control strategy). We will add some details about this choice in the paper. This can be considered as a "detuning strategy". This remark is important because we actually compare our strategy to a "detuning strategy" which already avoids the negative damping effects. We'll reformulate and add this to the text. However, the focus of the paper is on the floating platform compensation. Since the platform pitch control ($\beta_c omp = -k\beta\dot\phi$) has a damping effect on the frequencies close to the natural frequency of the platform, its tuning can reasonably be considered as independent from the tuning of the PI controller. This is why, in the paper, we didn't focus on the tuning of the PI controller, which is already well assessed in literature.

30. Section 3.3. The value of the left-hand side of the third inequality in (51) varied from case to case, but was usually between 0.055 and 0.065 (so indeed lower than 0.1). This is coherent with the fact that the system is highly under-damped.

31. Fig. 6. when the platform pitch rate is compensated, the Thrust changes because of the change in the blade pitch (induced by $k_\beta$) and the mean platform rotational position changes. In fact the system slightly shifts to another operating point.

32. Fig. 8 and 9. The main result shown here is that, while the mean values changes are very small, there are big changes in min-max and standard deviation. Figures 8 and 9 actually give more precision about this statement, since all the densities are plotted. Also DELs and extreme loads are analyzed later in the paper, on a more complete simulation.

33. Sec. 3.4. This choice is to show the as much results as possible. If necessary to be homogenized, it can be done.

34. Fig. 12. blade pitch saturation is not considered to better observe the effect of our platform pitch control strategy

35. Fig. 13. Max Rotor Speed and Max Gen Power will be added in the paper. the coupling affects rotor speed, hence, around $4m/s$ the speed is close to cut-in and the variation on the rotor speed can lead to values of power close to zero. However, it is negligible with respect to the global power production.

36. bibliography will be revised and improved

---

## Author Comment (AC3)

**Answers to reports on wes-2022-109**

Paul Mella and Matteo Capaldo

February 10, 2023

**Answer to anonymous referee 3**

1. ROSCO will be in the introduction. In particular, we'll shortly describe the platform pitch control strategy around line 64 and we'll cite the state-of-the-art also.

2. Line 46. The difference between the two papers is related to the fact that, formally, we don't change the rated speed. In the text, to be more clear, "as introduced in this paper" will be changed by "in (Lackner et al.)". We will also explain that by modifying the rated speed as function of the platform pitch is actually equivalent to adding a compensation term to the blade pitch control that depends on the platform pitch. However, it implies a coupling between the proportional and integral gains related to the platform pitch and those related to the rotor speed control. This is different from the work presented in this paper, where those parameters are independent (moreover we consider only a proportional gain for the platform pitch control, the integral one being 0).

3. Line 193. We'ill reformulate in the text. We propose to replace the sentence by "When the equation is verified, it means that: $\tau_a$ is more sensitive to blade pitch than rotational speed and $F_a$ is more sensitive to omega than blade pitch. Therefore, by increasing blade pitch $\omega$ increases and, then, occasions $F_a$ to decrease. Then, $\phi$ increases."

4. Line 220. We'll add a graphic visualization of the roots of G(s), we'll show how those roots move when $m_{\tau_g}$ varies, and that will explain how $k_{\tau_g}$ is chosen and used. We'll give more details about $m_{\tau_g}$, including its dependency on $D_t$. For this, the graphical representation of zeros and poles will be useful.

5. We'll revise the references to sections, figures, and tables.

6. Tables 1, 2, 3, Figures 2, 3, 4. display each table next to the associated figure. The values are determines arbitrarily as examples, to ensure the appearance of NMPZs. This is done in the purpose of a pedagogic example to show the effects of the NMPZs. The values are purely numerical. There are not so far from the coefficients related to the IEA15MW wind turbine because we started from those physical values and we found the right coefficients, in the order of 10% or 20%, to make the system unstable. The numerical section focuses on more realistic FOWT tests.

7. In section 3. (numerical simulations) the reference strategy (refered to as $k_\beta = 0$) corresponds to a "detuning strategy" (We will develop more about this when introducing the PI controller's parameters $\zeta_{rot}$ and $\nu_{rot}$) which already reduces the coupling effect between platform dynamics and rotor dynamics. The ROSCO toolbox, downloaded from github, implements this "detuning strategy" with interpolated gain coefficients $k_P$ and $k_I$ when the floating feedback is disactivated. For given turbine characteristics and operating point, one might choose wisely a fixed parameter $k_\beta$ (which corresponds to Fl_kp in the ROSCO toolbox), but the toolbox does not give indications on how to choose this parameter: instead, it suggests only one value of $k_\beta$ for all operating points and turbine characteristics, while we prove (see Figure 12.) that the appropriated $k_\beta$ depends on the wind speed. We think this value might be obtained by linearisation (by ROSCO toolbox), similarly to what is done for the gain scheduling of $k_P$ and $k_I$ (but independently to the wind speed). This method is similar to what is done in (Lenfest, 2020).

Moreover we noticed that the platform pitch controller strategy defined in the article (Abbas,

2022) (see lines 305-310, and especially equation (47) in our paper) is not implemented in the ROSCO toolbox. (Abbas, 2022) gives an explicit formula for $k_\beta$, but as we will explain in the next point (see below, point 8.) the obtained values for $k_\beta$ are very different (the sign is switched) from our formula. The results obtained by (Lenfest, 2020) are closer to our formula than the one given in (Abbas, 2022).

Concerning the simulations, choosing wisely (eg. with a linearisation by ROSCO toolbox), for a given operating point and turbine characteristics, a fixed parameter $k_\beta$ would give similar results to our strategy. The added value of our work is to be able to compute explicitly (without any calibration) the value of $k_\beta$ for any turbine characteristics and operating point, and to give an analytical support and an explicit formula corresponding to the numerical results already observed by (Lenfest, 2020). We consider that this question is very useful and we'll clarify this point in the introduction and in section 3.

8. Comparison about ROSCO platform pitch controller strategy and the proposed platform pitch controller strategy: The main difference in the two approaches can be remarked at lines $305-310$. As it can be remarked, the two ways to define the platform pitch compensation $k_\beta$. ROSCO derives the parameter imposing the rotor dynamics and the platform pitch dynamics to be decoupled at the first order. In other words, the effect of relative wind generated by the platform pitch dynamics is, at first order, compensated by feathering. The strategy proposed in this new paper aims at taking advantage of the blade pitch control influence on the platform pitch dynamics in order to introduce an extra term in the second order dynamics equation of the platform pitch. Thus the second order dynamics equation has an explicit form involving a damping ratio $\zeta_{plt}$ whose value one can explicitly define.

On can, then, notice that the two formulas to define $k_\beta$ for ROSCO strategy and the proposed strategy are different. Also numerically, they lead to values that are opposite in sign.

Indeed, $\frac{\partial F_a}{\partial \beta} < 0$ (for an above-rated operating point) and therefore using inequality (51), we find that

$$k_\beta = \frac{1}{h_t \frac{\partial F_a}{\partial \beta}} \left( 2\sqrt{K_t J_t}\zeta_{plt} - D_t - h_t^2 \frac{\partial F_a}{\partial v} \right) < 0$$

On the other hand, ROSCO strategy, as it is defined in (Abbas, 2022) or (Sotckhouse, 2021) derives from the equation $A_{2,4} = 0$ where $A$ is the matrix defined in (17) ((Abbas, 2022) introduces that same matrix) and expresses the platform pitch control coefficient as

$$k_\beta = -h_t \frac{\partial \tau_a}{\partial v} / \frac{\partial \tau_a}{\partial \beta} > 0$$

It is negative since $\frac{\partial \tau_a}{\partial v} > 0$ and $\frac{\partial \tau_a}{\partial \beta} < 0$ (for an above-rated operating point). Notice that in (Abbas, 2022), $\beta_{comp}$ is defined as in (Stockhouse, 2021) but with the convention $\beta_{comp} = k_{float}\dot{\phi}$ so that $k_{float} = -k_\beta = h_t \frac{\partial \tau_a}{\partial v} / \frac{\partial \tau_a}{\partial \beta}$ is negative, but this is just a question of conventions. If one takes the same convention, the sign is actually switched in our formula.

9. Line 254. This sentence is proposed: "It is complicated to explicit[ly determine] the damping"

10. Line 374. This sentence is proposed: "which was analysed at first order in [section] 2.5"

11. we should add a paragraph explain hox $k_\beta$ is defined in ROSCO (right after 2.5.2 for example) and explain why the sign is switched.

12. For test cases in section 3.3, control signals are partially reported (rotor speed). The blade pitch can be added in the text. If this is interesting, we propose to report rotor speed and blade pitch in annex.

For numerical tests in section 3.4, we can produce the the control signals, generator speed, and platform pitch for the proposed controller. However, we suggest to send you those figure in the discussion without reporting them in the paper. They would not add any further information and, since the wind is turbulent, they will be not easy to be interpreted. Alternatively, it can be done in an annex.

13. About the wind energy verbiage: We'll do our best to improve the verbiage and adapt it to the wind energy audience. However, "wave period" is, for instance, a typical way in offshore wind to indicate the period (inverse of frequency) of the incoming waves.

14. answer to Figure 10: Section 2.5.2 shows how the proposed strategy add an extra damping in the platform pitch by coupling rotor dynamics with platform pitch dynamics. It leads to reduce the platform pitch dynamics, however, it leads also to variation in the rotor speed. There are references to this effect at line 375 and 405.

15. Line 370. "diagram 5" will be changed to "Figure 5".

16. Figures 8 and 9 report Tower base moment (load on tower), where Figure 6 and 7 report platform pitch. The idea is to show that reducing the platform movements it will reduce the tower base moment. This is something one can imagine but it is interesting to show it by results.

17. Quality factor is defined by
$$Q = \frac{1}{2\zeta}$$

As some readers might be more used to work with quality factors instead of damping ratios, we thought it was a good thing to give the quality factors corresponding to the damping ratios. It is not mandatory for the comprehension of the paper. If you prefer, we could just delete the sentence about the quality factor, or just put it in a foot note (linked to the previous sentence).

---

## Author Response (AR2)

**Answers to reports on wes-2022-109**

Paul Mella and Matteo Capaldo

April 26, 2023

**Answer to anonymous referee 1**

1. Reviewer claims: "My earlier question on Figure 6 "Why does the mean ptfm pitch change, if we're just feeding back velocity, rather than position?" is not resolved. I did not compare against Figure 5 but only the lines within Figure 6. I don't understand why the mean value would change in closed-loop if the "plt" controller feeds back velocity rather than position."

   We are sorry that we didn't understand the question last time. Now it is clear. The change in the mean platform pitch value is motivated by the difference in the mean value of the blade pitch (see Figure 1 of this document). One can compare the high values in the peaks for $\zeta_plt = 0.25$ with respect to the other curves. The minimum does not change with respect to the other cases. It results in a higher mean value for the blade pitch. Thus, the latter leads to a lower thrust force in average. This makes the platform pitch mean (the static part) value lower than $\zeta_plt = 0.1$ and the reference.

   The reason of the peaks in the blade pitch can be motivated by the combination of high demanded damping ($\zeta_plt = 0.25$), with the proximity to the rated wind speed (the controller is on the boundary of regions 2.5 and 3) and also the wave period. In fact, for case (2), where the wind speed is the same, the wave period is much higher and this phenomenon is not produced (the platform pitch mean value is the same for all the values of $\zeta_plt$).

2. Remark accepted and integrated in the manuscript.

3. Remark accepted and integrated in the manuscript.

[Figure]

Figure 1: Blade pitch output for simulation reported in Figure 9 of manuscript (test case (1)). Mean value of the blade pitch for $\zeta_{plt} = 0.25$ is higher the other curves.

**Answer to anonymous referee 2**

1. Reviewer claims: "It is still difficult to understand the merits of this controller, compared to the standard ROSCO controller with a flipped sign $k_\beta$. The authors maintain that only a de-tuning method is used in ROSCO, but that is not the case as is openly defined here: `https://github.com/NREL/ROSCO/blob/main/Test_Cases/IEA-15-240-RWT-UMaineSemi/DISCON-UMaineSemi.IN`.

   This issue has been extensively addressed in the previous answer to the reviewers. In the manuscript, there is an entire Section (Section 2.5) dedicated to the the differences with respect to ROSCO controller. In this Section, equations (46) reports the explicit expression of $k_\beta$ for our control strategy. equations (47) reports the explicit expression of $k_\beta$ for ROSCO strategy. The two expressions are different because ROSCO calibrates the $k_\beta$ in order to decouple the platform pitch dynamics and the rotor dynamics, while our strategy focuses on the platform damping. Moreover, The ROSCO controller, as downloaded by the github link, uses a unique $k_\beta$ for a given simulation, even if the wind speed varies. In our strategy the $k_\beta$ is explicitly defined and it is implemented to variate in the time series. With this paper, the reader is provided with an explicit formulation of the platform pitch compensation allowing to increase the platform damping and this explicit formulation has never been reported in literature. This is an important novelty (not present in ROSCO controller or papers related to ROSCO). Finally, it is to be underlined that we have never mentioned that only a de-tuning method is used in ROSCO. Perhaps the reviewer got confused because, for numerical tests, in Section 3, we have chosen a de-tuning strategy as reference for the comparison of results.

2. Thanks, it is corrected in the new version of the manuscript.

---

## Author Response (AR3)

**Answers to editor on wes-2022-109**

Paul Mella and Matteo Capaldo

May 27, 2023

**Answer to the editor**

1. The editor demanded to "include the defined controller in ROSCO (including the constant $k_\beta$, rather than detuning) for this platform". For this, authors have added in Section 3.4 a further term of comparison which considers a constant and unique value of $k_\beta$. The value is chosen to be the one proposed by ROSCO for this FOWT system (see https://github.com/NREL/ROSCO/blob/main/Test_Cases/IEA-15-240-RWT-UMaineSemi/DISCON-UMaineSemi.IN).

   During the code implementation of this point, one mistake has been discovered for the image showing the DEL of the bearing of the blade pitch. The error concerned only the wording "increment" (results were correct). This word "increment" was inappropriate because the value reported are related to a ratio between the the different strategies. Hence, an increment lower than 100% results in a decrease of the fatigue of the bearing. It induced in error also the authors which commented in accord with the word "increment". This word is corrected by "ratio" and the comments to those results are corrected also.

2. The editor demanded: "Section 3.4 should include some time series to illustrate the effects of the different controllers on control signals, generator speed, and platform pitch. These can be short time series for a single case, but are needed to increase understanding." Considering the wind condition 10 $ms^{-1}$, time-series of the quantities of interest have been extracted from the simulation. 150 seconds plot are reported. A larger time would make the images hard to be read.

3. An English native-speaker has reviewed the paper. Many improvements are introduced for the English and some extraneous words are identified and replaced.

---

## Author Response (AR4)

**Answers to editor on wes-2022-109**

Paul Mella and Matteo Capaldo

June 17, 2023

**Answer to the editor**

The paper is modified according to the suggestions of the editor (details in the next of this document). Moreover, the paper is generally improved and in particular the section 3. An error is corrected in the computing of the fatigue of the pitch bearing (Figure and comments are corrected).

The editor remarked:

1. "Thank you for the revisions. I would still request one last round of revision related to Section 3. The results added to Section 3 generally strengthen the paper, but Section 3 has become rather difficult to follow. It is suggested that a table should be added to more clearly define all of the different control strategies that are compared, and the strategies should be referred to with names that are easy to relate back to such a table".

   Answer: Thank you for the remarks and corrections. The suggestion is accepted and a table is added resuming the controller strategies compared with the respective parameters $k_P, k_I and k_\beta$. The names used for the strategies in the text are the same employed in this table, helping the reader to have a reference of the strategies compared.

2. "Line 49: Note that the blade pitch control responds to an increase in rotor speed, not directly to the increase in the relative wind speed."

   Answer: the link between wind speed increase, aerodynamic torque increase and rotor acceleration was implied for brevity. A longer sentence is now in the manuscript to clarify this mechanism.

3. "Line 151: operative or operational?"

   Answer: "operational" seems more adapted in this sentence to express the idea that the compensation is used to obtain the same performance of a bottom-fixed wind turbine. Hence "operational" replaces "operative" in the manuscript.

4. "Line 158: typo "ot", and "it translates to" rather than "in""

   Answer: both corrected, thanks.

5. "Line 190: The sentence beginning "every component" is not a grammatically complete sentence."

   Answer: Sentence corrected with the following one: "Every component of  G(s) can be written as the quotient of a polynomial in $s$ and $\chi_a(s)$".

6. "Figures 6 and 7 should be include axis labels and Figure 7 should be explicitly named in the text."

   Answer: Figure 7 is now explicitly named in the text. Values in Figure 6 and 7 are poles and zeros of the transfer function of the system in the complex domain. The labels for the real and imaginary part of the complex domain are now added in the manuscript.

7. "The explanation for the change in mean platform pitch in Figure 9 should be included in the paper."

   Answer: The answer given to the reviewer n.1 is now added in the manuscript with the associated Figure.

8. "Why not include the ROSCO controller in these results with regular waves and constant wind as well? "

Answer: The test case with constant wind and monochromatic waves aims at verifying the analytical developments of the previous section. For this test, it is not recommended to compare performances of different control: it has not an added value in the comparison of the different controller strategy because those are ideal environmental conditions, they are not realistic and any comparison would not conclude on the pertinence of the controller strategy. In fact a controller performing well in this case could perform badly perform in a more realistic environment. The test has a sense when an analytical form of the controller is given and the comparison with the analytical results can be given. The idea is to show that a numerical model evolving in the same conditions of the analytical analysis reproduces the expected behaviour.

9. "And are detuned coefficients used with the ROSCO controller? This seems unrealistic, as the aim should be to avoid detuning."

Answer: Concerning the coefficients $\nu_{rot}$ and $\zeta_{rot}$, many tests have been performed to select the ones giving the best results for this floating wind turbine. Once tuned, they are kept constant for the three strategies. It is to be underlined that in ROSCO controller the tuning of $k_\beta$ is not correlated to the tuning of $\nu_{rot}$ and $\zeta_{rot}$. In fact, those coefficients are tuned separately from the floating feedback coefficient, as done in this article. Moreover "detuning" is actually the adaptation of the coefficient to the specific floating system (see Hu, et al. (2021), Implementation and evaluation of control strategies based on an open controller for a 10 MW floating wind turbine. Renewable Energy. 179. 10.1016/j.renene.2021.07.117.).

As the editor underlined previously, it is worth to show the benefit of considering an explicit definition of $k_\beta$ adapting to the operating point and evolving in time with respect to a single time-constant and not adapting $k_\beta$. It doesn't depend on the choice of $\nu_{rot}$ and $\zeta_{rot}$. What is important is that the same $\nu_{rot}$ and $\zeta_{rot}$ are considered for the different strategies, in order to underline the benefit of the choice of $k_\beta$. A comparison considering too many parameters changing from one term of comparison to another one is difficult to be understood.

10. Line 588: "it reduces fatigue damage" by, not for

Answer: corrected